# Blocking the GITR-GITRL pathway to overcome resistance to therapy in sarcomatoid malignant pleural mesothelioma

Meilin Chan [1,2,3,4], Licun Wu[2], Zhihong Yun[2], Trevor D. McKee [5], Michael Cabanero[6], Yidan Zhao[2], Mikihiro Kohno[1,2], Junichi Murakami [1,2] & Marc de Perrot[1,2,7 ✉]

Malignant pleural mesothelioma (MPM) is an aggressive neoplasm originating from the pleura. Non-epithelioid (biphasic and sarcomatoid) MPM are particularly resistant to therapy. We investigated the role of the GITR-GITRL pathway in mediating the resistance to therapy. We found that GITR and GITRL expressions were higher in the sarcomatoid cell line (CRL5946) than in non-sarcomatoid cell lines (CRL5915 and CRL5820), and that cisplatin and Cs-137 irradiation increased GITR and GITRL expressions on tumor cells. Transcriptome analysis demonstrated that the GITR-GITRL pathway was promoting tumor growth and inhibiting cell apoptosis. Furthermore, GITR+ and GITRL+ cells demonstrated increased spheroid formation in vitro and in vivo. Using patient derived xenografts (PDXs), we demonstrated that anti-GITR neutralizing antibodies attenuated tumor growth in sarcomatoid PDX mice. Tumor immunostaining demonstrated higher levels of GITR and GITRL expressions in non-epithelioid compared to epithelioid tumors. Among 73 patients uniformly treated with accelerated radiation therapy followed by surgery, the intensity of GITR expression after radiation negatively correlated with survival in non-epithelioid MPM patients. In conclusion, the GITR-GITRL pathway is an important mechanism of autocrine proliferation in sarcomatoid mesothelioma, associated with tumor stemness and resistance to therapy. Blocking the GITR-GITRL pathway could be a new therapeutic target for non-epithelioid mesothelioma.

[1] Division of Thoracic Surgery, Toronto General Hospital and Princess Margaret Cancer Centre, University Health Network, Toronto, ON, Canada. [2] Latner Thoracic Surgery Laboratories, Toronto General Hospital Research Institute, University Health Network, Toronto, ON, Canada. [3] Mackay Memorial Hospital, Taipei, Taiwan. [4] Institute of Traditional Medicine, School of Medicine, National Yang Ming Chiao Tung University, Taipei, Taiwan. [5] The STTARR Facility, Toronto General Hospital Research Institute, University Health Network, Toronto, ON, Canada. [6] Department of Pathology, University Health Network, Toronto, ON, Canada. [7] Department of Immunology, University of Toronto, Toronto, ON, Canada. ✉email: marc.deperrot@uhn.ca

Malignant pleural mesothelioma (MPM) is a rare and fatal malignancy of the pleura linked to occupational and environmental asbestos exposure[1]. Recent studies demonstrated that the overall median survival remains poor despite multimodality therapy in MPM[2,3]. A novel therapeutic approach with Surgery for Mesothelioma After Radiation Therapy (SMART) demonstrated encouraging outcome in epithelioid MPM with a median survival of 36 months[4]. However, patients with nonepithelioid subtypes still carried poor prognoses with a median survival of 14 months. It is well established that the sarcomatoid and biphasic mesothelioma subtypes are refractory to conventional therapies including surgery, chemotherapy, and radiotherapy with median survivals of less than 12 months in large trials[5,6]. Thus, identifying the mechanisms of resistance in nonepithelioid mesothelioma could potentially provide a major step forward for developing more effective therapies in this disease.

Evidence shows that a small population of cancer cells, which are considered cancer stem cells, contains higher tumorigenicity and resistance to chemotherapy and radiotherapy that may lead to the repopulation of malignant cancer cells after therapy[7]. In our previous work, we discovered that the expression of Glucocorticoid-Induced TNFR-Related Protein Ligand (GITRL, Tnfsf18) could be induced by radiation and chemotherapy and be a marker of stem-like cells associated with resistance to chemotherapy or radiotherapy in murine mesothelioma[8]. Like many members of membrane-bound TNF family, GITRL is expressed by immune cells and interact with its cognate receptor (GITR), generating downstream as well as reverse signaling[9]. In murine tumor models, the engagement of GITR by anti-GITR agonist has been shown to improve survival and even eradicate the tumor by activating an immune response[10,11]. The concept of immunosurveillance suggests that the development of clinically apparent cancer depends on the ability of cancer cells to escape the immune system through immunoediting[12]. Many tumor necrosis factors (TNF)/TNFR family influence host-tumor interactions by governing differentiation, proliferation, activation, and death of both tumor and immune effector cells[13]. Hence, expression of GITR/GITRL may be a mechanism of immune escape developed by cancer cells. In chronic lymphocytic leukemia (CLL), GITR/GITRL interaction between GITRL-expressing cancer cells and GITR-expressing NK cells reduced the cytotoxicity and IFN-γ production of the NK cells, thus allowing the immune cells to escape immunosurveillance. Additionally, GITR/GITRL interaction can induce cancer cells to release IL-6, IL-8, and TNF, which then act as growth factors for CLL cells and cause an autocrine loop[14]. The role of GITR/GITRL on cancer cells has not been investigated in solid tumors. In this study, we therefore aimed to elucidate the role of GITR/GITRL as a potential mechanism of resistance to therapy in mesothelioma.

## Results

**GITR/GITRL expression is associated with resistance to chemo and radiotherapy in human mesothelioma cell lines**. We first clarified the histological characteristics of three human mesothelioma cell lines CRL5820, CRL5915, and CRL5946 in vitro and in vivo in NOD/SCID mice. We found that CRL5820 cell line was epithelioid, CRL5915 biphasic, and CRL5946 sarcomatoid (Fig. 1a). Using molecular markers for mesenchymal characteristics (calretinin, N-cadherin, β-catenin, vimentin, Slug, and Snail), we confirmed that CRL5946 and CRL5915 expressed mesenchymal markers supporting their sarcomatoid component. Our previous work showed that GITRL expression was related to stemness in murine RN5 mesothelioma cell lines[8]. We therefore analyzed the transcript and protein expression of GITR and

GITRL in the three human cell lines CRL5820, CRL5915, and CRL5946 by measuring their mRNA and protein expression levels. Quantified RT-qPCR showed that GITRL-gene expression was significantly increased in CRL5946 cells, while GITR-gene expression was similar between the three mesothelioma cell lines (Fig. 1b, Supplementary Data 1). Immunoblot analysis of total lysates confirmed that CRL5946 cells highly expressed both GITR and GITRL (Fig. 1c). In CRL5946, dose-dependent cytotoxicity test for cisplatin and Cs-137 irradiation showed that 2 μg/ml of cisplatin and 7.5 Gy of Cs-137 irradiation caused 70–80% cell death (Fig. S1). Thus, we used these dosages to treat mesothelioma cell lines in further experiments. The proliferation rate was higher in CRL5946 than in CRL5820 or CRL5915 at baseline (Fig. 1d, Supplementary Data 1). After treatment with 2 μg/mL cisplatin or 7.5 Gy Cs-137 irradiation, the proliferation rate was decreased across all three cells lines, but remained similarly superior in CRL5946 (Fig. 1d). In addition, after treatment with the same dosage of cisplatin or Cs-137 irradiation, we found more colony formation in CRL5946 than in CRL5820 or CRL5915 (Fig. 1e). Overall, these results showed that the sarcomatoid mesothelioma cell line (CRL5946 cells) constitutively expressed GITR and GITRL and was associated with greater proliferation rate, chemoresistance, and radioresistance than epithelioid (CRL5820) and biphasic (CRL5915) cell lines. Therefore, these findings suggest that GITR and GITRL may play a role in the mechanisms of progression of sarcomatoid mesothelioma and resistance to conventional therapy.

**Cisplatin and Cs-137 irradiation induces GITR and GITRL expression in sarcomatoid mesothelioma cells (CRL5946)**. We next examined whether cisplatin or Cs-137 irradiation is capable of inducing GITRL or GITR expression in sarcomatoid mesothelioma cells. The expression of GITR and GITRL on CRL5946 cells continuously increased over time after chemotherapy and radiotherapy (Fig. S2). GITRL expression peaked on the 4th day after radiation and chemotherapy, while GITR expression peaked on the 4th day after chemotherapy and continued to rise up to 7 days after radiation. We collected CRL5946 cells 4 days after treatment with cisplatin (2 μg/mL) or with Cs-137 irradiation (7.5 Gy) to detect gene expression and protein level of both GITR and GITRL. Using RT-qPCR, we found that cisplatin or Cs-137 irradiation significantly increased GITR and GITRL expression compared to no treatment (Fig. 2a, Supplementary Data 1). Immunoblot analysis (Fig. 2b) and immunofluorescence (Fig. 2c) also showed that GITR and GITRL expression were increased in cisplatin- and Cs-137-treated group compared to the untreated group. Note that the time of fluorography for capturing the images was shorter in Fig. 2b compared to Fig. 1c, which explain the weak appearance in the untreated group. Taken together, our results demonstrate that chemo- and radiotherapy increase GITR and GITRL expression in sarcomatoid mesothelioma cells (CRL5946). Cisplatin and Cs-137 irradiation also increased GITR and GITRL expression in epithelioid (CRL5820) and biphasic (CRL5915) mesothelioma cells (Fig. S3).

**Increase of disease and biological hallmarks in GITR+ and GITRL+ cells**. Our data revealed that there are three distinct GITR/GITRL subpopulations within the CRL5946 cells: GITR+ GITRL−, GITR−GITRL+, and GITR−GITRL− (Fig. 3a). To further investigate the mechanisms of GITR- and GITRL-related signals, the three populations were separated using the magnetic bead isolation kit, and their purity was verified by flow cytometry (Fig. 3a). We conducted the microarray analysis to profile the transcriptomes of the three cell populations (Supplementary Data 2). The principal component analysis showed that major

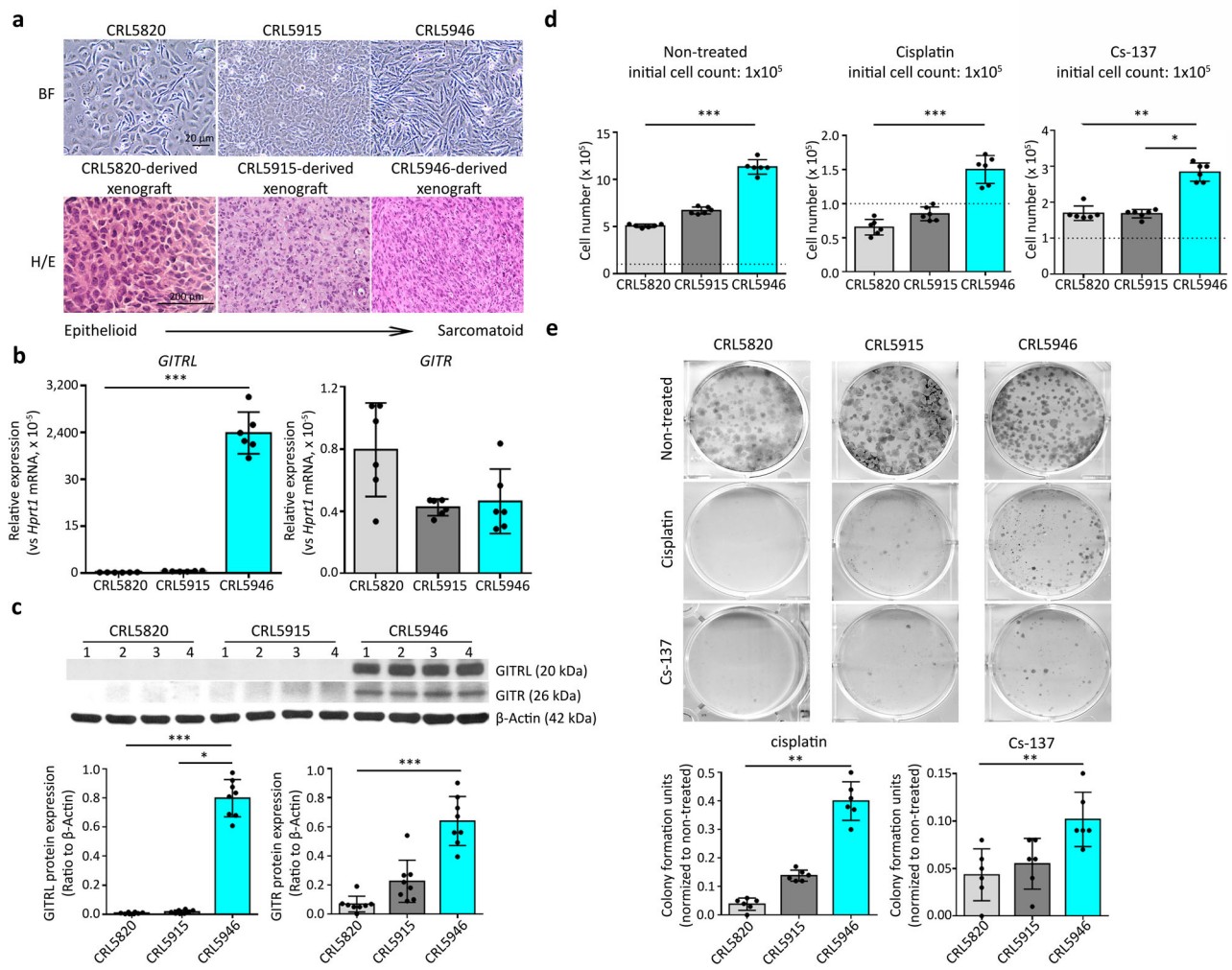

**Fig. 1 Characteristics of three human mesothelioma cell lines. a** Top panel: Representative image of phase contrast microscopy shows the difference of cell morphology among the three cell lines. Bottom panel: A total of 6 × 10⁶ of CRL5820, CRL5915, or CRL5946 cells were subcutaneously or intraperitoneally injected into NOD/SCID mice. Representative images of H&E staining of xenografted tumor from mice in each treatment group. **b, c** Comparison of GITRL and GITR expression among mesothelioma cell lines by using qPCR (**b**) and immunoblot analysis (**c**) (*n* = 8). **d** Mesothelioma cell lines were treated with 2 μg/mL cisplatin or exposed to 7.5 Gy Cs-137 irradiation and the cell proliferation was analyzed by counting the number of viable cells 4 days after treatment (*n* = 6). **e** Representative image of colony formation assay and quantification of visualized colonies in cells 14 days after exposure to 2 μg/mL cisplatin or Cs-137 7.5 Gy irradiation (*n* = 6). Data were analyzed with non-parametric Kruskal–Wallis test followed by the Mann–Whitney U test with Dunns correction. (**P* < 0.05, ***P* < 0.01, ****P* < 0.001).

transcriptomic differences were found between GITR−GITRL− cells and GITR+ or GITRL+ cells. However, no distinct clustering was observed between GITR+ cells and GITRL+ cells (Fig. 3b). We imported the different genes expressed between GITR−GITR− population and GITRL+ or GITR+ populations with *P* value less than 0.05 and FDP value less than 0.1 into Ingenuity Pathway Analysis (IPA). The canonical pathway analysis revealed that the dolichyl-diphosphooligosaccharide biosynthesis [−log(*P* value) = 4.06, Z-score = 2], UVC-induced MAPK signaling [−log(*P* value) = 2.24, Z-score = 1.34], neuregulin signaling [−log(*P* value) = 2.2, Z-score = 1.34], Huntington's signaling [−log(*P* value) = 1.44, Z-score = 1.63], and EIF2 signaling [−log (*P* value) = 1.35, Z-score = 1.34] were upregulated in the GITRL+ cells, while sumoylation pathway [−log (*P* value) = 2.13, Z-score = −2.45] was down-regulated. In the analysis of GITR+ cells, the cell-cycle control (estrogen-mediated S-phase entry [−log(*P* value) = 3.96, Z-score = 1.34], cyclins and cell-cycle regulation [−log(*P* value) = 3.08, Z-score = 1.13]), NER pathway [−log (*P* value) = 2.48, Z-score = 1.13], and oxidative phosphorylation [−log(*P* value) = 1.77,

Z-score = 2.45] were upregulated. On the contrary, downregulation of cell-cycle checkpoint control (role of CHK proteins in cell-cycle checkpoint control [−log(*P* value) = 5.96, Z-score = −1.41], G1/S checkpoint regulation [−log(*P* value) = 2.07, Z-score = −1.34]), and p53 signaling [−log (*P* value) = 2.26, Z-score = −2.23] and p70S6K Signaling [−log (*P* value) = 2.3, Z-score = −1.13]) were observed (Fig. 3c).

IPA disease and functional analysis pointed out that the most significantly enriched diseases and biological function [−log (*P* value) >1.3, Z-score>2 or <−2] of GITR+ and GITRL+ cells were associated with cancer, upregulated cell-cycle regulation (G2/M phase, S phase of tumor cell lines) and colony formation (only in GITRL+ cells). On the contrary, function of cellular death, apoptosis, and development of cytoplasm (only in GITR+ cells) were decreased (Fig. 3d). Taken together, these analyses suggest that mesothelioma cells with GITR and GITRL expression proliferate more than GITR − GITRL− mesothelioma cells and are less susceptible to death through inhibition cell apoptosis by activation of various signaling pathways.

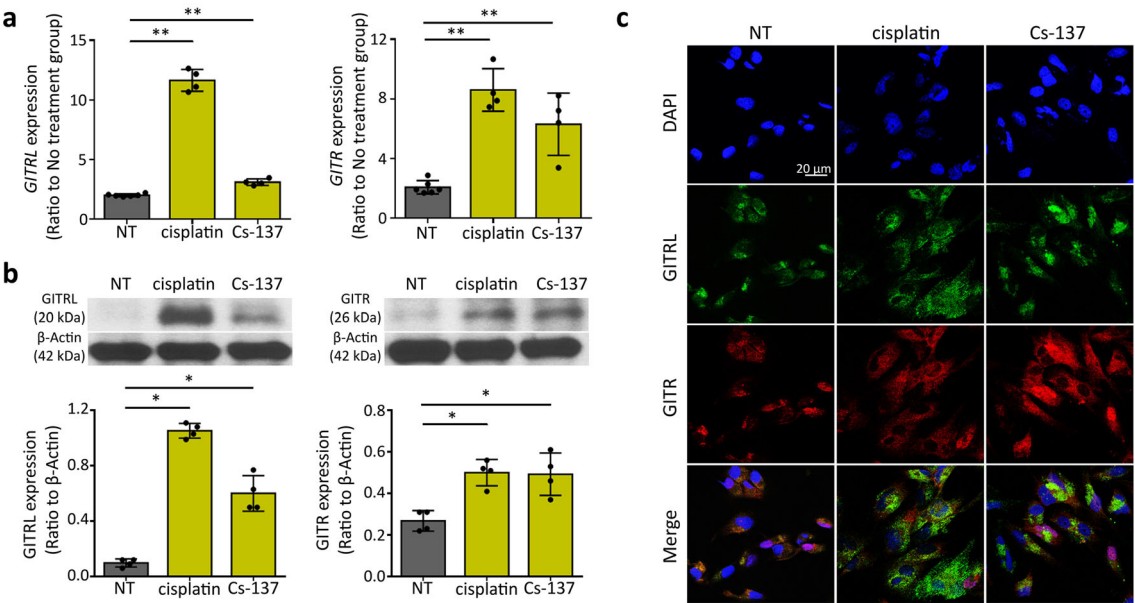

**Fig. 2 Expression of GITRL and GITR in sarcomatoid mesothelioma cell line (CRL5946) increased when treated with cisplatin or Cs-137 irradiation.** CRL5946 cells were exposed to cisplatin 2 μg/ml for 24 h or Cs-137 irradiation 7.5 Gy then collected 4 days later. **a** qPCR shows that GITRL and GITR mRNA expressions are significantly increased in cisplatin-treated and Cs-137-irradiated cells. **b** Representative histograms show GITRL and GLTR protein expressions are increased in cisplatin-treated and Cs-137-irradiated cells. **c** Immunofluorescence staining for GITRL (green), GITR (red), DAPI (blue) in CRL5946 treated with cisplatin or Cs-137 irradiation. DAPI was used as a nuclear marker. *$P < 0.05$; **$P < 0.01$; vs nontreated control, determined by using Mann–Whitney U test.

**GITR+ and GITRL+ CRL5946 cells have increased tumor-igenicity in xenografted mice.** Since the transcriptomic profiles suggested that GITR and GITRL expression were associated with cell proliferation and tumor formation, we used neutralizing anti-GITR mAb to examine the function of GITR and GITRL in cell proliferation and tumorigenicity in vitro and in vivo. The neutralizing anti-GITR mAb (N-mAb) that we used is an antagonistic anti-GITR antibody, which prevents GITRL from binding to GITR and thus blocks both GITR and GITRL-mediated signals[15].

We purified GITR+ and GITRL+ cells from CRL5946 cells by using the magnetic beads isolation kits. Purified cells were grown with serum-free, growth factor-rich medium in an ultralow attached plate for 14 days (Fig. 4a). We observed more formation of spheroids with size over 50 μm in diameter in GITR+ and GITRL+ group than in GITRL–GITR− group in vitro (Fig. 4b, left panel, Supplementary Data 1). For in vivo investigation, purified cells were intraperitoneally injected into NOD/SCID mice for 30 days (Fig. 4a). We found that there was more formation of spheroids with size over 70 μm in peritoneal lavage in GITR+ and GITRL+-injected mice than in GITR − GITRL −-injected mice (Fig. 4b, right panel). Moreover, when culturing those isolated subpopulations as a monolayer fashion on plates, the distribution of these three subpopulations returned to normal equilibria demonstrating that there is a stochastic interconversion between GITR−GITRL−, GITR+, and GITRL+ cells contributing to the heterogenicity of CRL5946 mesothelioma cells (Fig. S4).

We used the N-mAb to further demonstrate the requirement of GITR/GITRL-mediated signaling for cell proliferation in vitro and tumorigenesis in vivo. Since the expression of GITR and GITRL in CRL5946 mesothelioma cells can be induced by exposing them to cisplatin or Cs-137 irradiation, we treated CRL5946 cells with 2 μg/mL cisplatin or 7.5 Gy Cs-137 irradiation and then blocked GITR/GITRL-mediated signaling by using N-mAb for 3 days (Fig. 4c). We found that N-mAb decreased cell proliferation at a high concentration of 80 μg/mL in nontreated cells and at low concentration of 20 μg/mL in cisplatin-treated or Cs-137-irradiated cells (Fig. 4d). Furthermore, we intraperitone-ally injected cisplatin-treated or Cs-137-irradiated cells into NOD/SCID mice for 24 days. Tumor-bearing mice were then given PBS, IgG isotype (100 μg), or N-mAb (100 μg) on day 0 and Day 7 (Fig. 4c). Compared to IgG isotype control, N-mAb significantly decreased intraperitoneal spheroid (>70 μm) formation in untreated cells, cisplatin-treated cells, and Cs-137-irradiated cells (Fig. 4d). The inhibition generated by N-mAb was more pronounced in cisplatin-treated cells and Cs-137-irradiated cells than in untreated cells with a 55% reduction in spheroid formation in cisplatin-treated cells, 41% reduction in Cs-137-irradiated cells, and 33% in untreated cells. These results demonstrate that GITR/GITRL-mediated signaling pathway contributes to cell proliferation and tumorigenicity in mice.

**Interruption of GITR/GITRL axis inhibits tumor growth in sarcomatoid patient-derived xenografts (PDXs).** We next examined the role of GITR/GITRL axis in tumor growth in patient-derived xenografted mice. Two patient-derived xenografts (PDXs), epithelioid subtype from patient#24 (tumor#24) and sarcomatoid subtype from patient#17 (tumor#17), were generated directly from MPM patients and confirmed histologically after passaged for seven generations[16]. We subcutaneously implanted 1–2 mm pieces of PDXs into NOD/SCID mice on the right flank. Mice with xenografted tumor size that grew up to 30–70 mm³ were intraperitoneally given PBS, 5 mg/kg cisplatin, or 400 μg N-mAb on day 0 and PBS, 5 mg/kg cisplatin or 200 μg anti-GITR mAb on day 7 (Fig. 5a). Immunohistochemistry staining showed that GITRL expression was markedly increased in sarcomatoid PDX compared to epithelioid PDX (Fig. 5b). As expected, we found that cisplatin significantly inhibited tumor growth in epi-thelioid and sarcomatoid PDX-implanted mice. However, we observed that the N-mAb significantly suppressed tumor growth

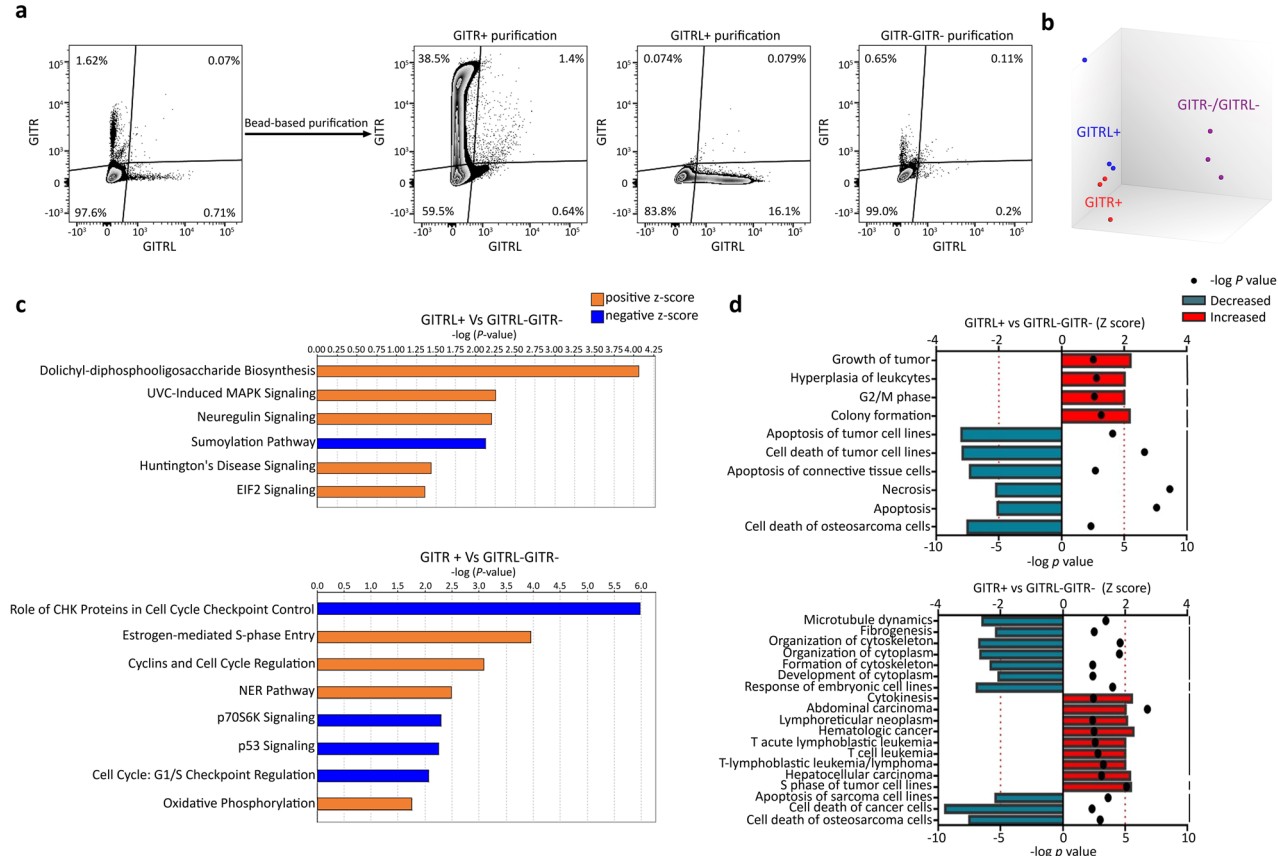

**Fig. 3 Microarray transcriptome analysis of GITR and GITRL signaling in CRL5946 cells. a** Flow cytometry analysis of GITRL+ or GITR+ cells sorted by anti-GITRL or anti-GITR antibody. Three subpopulations of CRL5946 cells, GITRL+, GITR+, or GITR−GITRL− cells, were separated by using the magnetic beads isolation kits. The separated cells were stained with anti-GITRL or anti-GITR antibody and analyzed by flow cytometry. The transcriptome profiling in three subpopulations of CRL5946 cells was analyzed by microarray. **b** Principal component analysis compared the transcriptome profile of GITRL+, GITR+, or to double negative cells. **c, d** IPA comparison of GITRL+ or GITR+ with double negative cells by uploading the significantly different expression of upregulated and downregulated genes (p < 0.05, FDP < 0.1) revealed predicted up- or downregulated canonical pathways (Z-score > 1, orange or < −1, blue). **c** and predicted disease and biological function analysis. **d** (Z-score > 2, red; <−2, cyan). Z-score represents the IPA trends (Z-score > 1, upregulated; Z-score < −1, downregulated). Results are evaluated by the negative log P value greater than 1.3.

in sarcomatoid PDXs, but not in epithelioid PDXs (Fig. 5c, Fig. S5, Supplementary Data 1). These findings confirm that the GITRL/GITR-mediated pathway is important for cellular proliferation in sarcomatoid mesothelioma with high GITRL and GITR expression.

**Elevation of GITR and GITRL expression in nonepithelioid mesothelioma patients correlates with worse survival.** To understand the impact of GITR and GITRL on survival, we examined their expression levels in 117 MPM patients (85 epithelioid, 29 biphasic, 2 sarcomatoid, 1 desmoplastic) treated in our institution from 2003 to 2016. The majority of patients (86%) were pretreated with induction chemotherapy or high dose hypofractionated hemithoracic radiation therapy as part of our innovative approach of Surgery for Mesothelioma After Radiation Therapy (SMART) before the measurement of GITR and GITRL on the surgical specimen (Fig. S6). The Kaplan–Meier survival analysis of all 117 cases demonstrated a median survival of 28.6 months in epithelioid subtype and 15.9 months in none-pithelioid subtype (Fig. 6a, Supplementary Data 1). Immunohistochemistry staining showed that the average intensity of GITRL and GITR were higher in nonepithelioid subtype than in epithelioid subtype, and that the percentage of GITRL was higher in the nonepithelioid subtype (Fig. 6b, Fig. S7). GITR levels

positively correlated with GITRL levels (Fig. 6c). To analyze the impact of GITR and GITRL levels on survival, we included only patients treated with the SMART protocol to homogenize the therapeutic approach as part of a prospective clinical trial (NCT00797719). In the risk analysis model of patients with nonepithelioid subtype, the survival negatively correlated with GITR levels, while in patients with epithelioid subtype, the survival positively correlated with GITRL levels (Fig. 6d, Supplementary Data 1). By using the median survival time, we separated patients into good prognosis and poor prognosis subgroup in both epithelioid and nonepithelioid subtype. We plotted the ROC (Receiver–Operator Characteristic) curve to determine the predictability of survival by using GITRL/GITR expression intensity. In epithelioid subtype, the GITRL level (AUC: 0.646, P = 0.08) and in nonepithelioid subtype, both the GITR and GITRL (AUC:0.77, P = 0.06; AUC:0.83, P = 0.02, respectively) could predict survival (Fig. S8). The determination of the ideal threshold of GITR/GITRL in both subtypes was defined according to the highest likelihood ratio. Based on the ideal threshold value of both GITR and GITRL, we classified patients into high or low expression of GITR/GITRL. The Kaplan–Meier survival analysis showed that in patients with nonepithelioid subtype, the high expression of GITR was associated with worse survival (Fig. 6e). However, in patients with epithelioid subtype, high

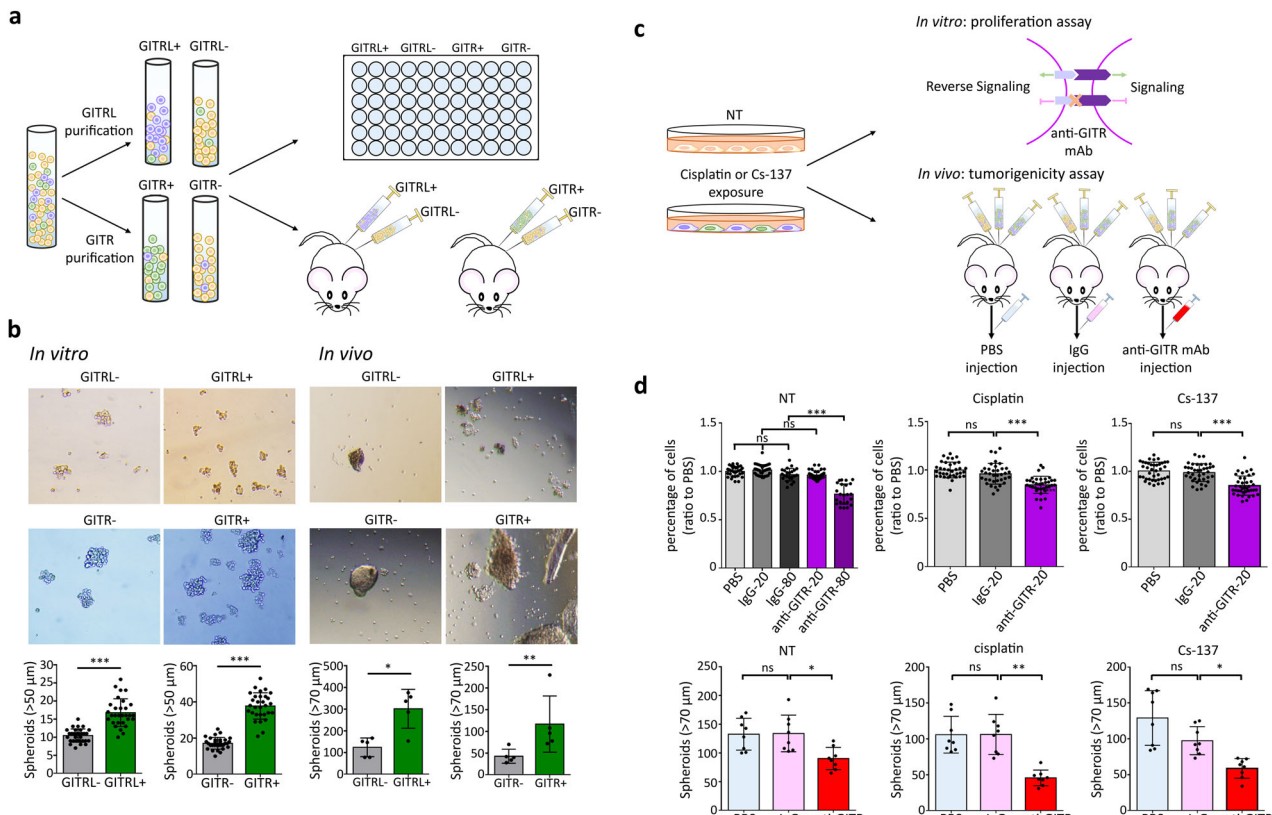

**Fig. 4 GITR/GITRL signaling contributes to tumorigenicity and cell proliferation in CRL5946 mesothelioma cells. a** Schematic representation of the experimental design and analysis of sphere formation from purified GITRL+ or GITR+ cells from CRL5946 cells in vitro and in vivo. **b** Representative images of in vitro cells that formed mesospheres (>50 μm) and quantification of spheroid growth from the purified GITRL+, GITRL−, GITR+, or GITR− populations in ultralow attachment plate for 14 days (left panel, n = 30 for each group). The purified cells were intraperitoneally injected into NOD/SCID mice to grow mesospheres. Representative images of spheres were taken on day 30 after cell injection. The size of mesospheres over 70 μm has been quantified to compare the tumorigenicity between GITRL+/GITRL− or GITR+/GITR− (right panel, n = 6 for each group). **c** Schematic representation of the experimental design by using the GITR neutralizing monoclonal antibody to verify the requirement of GITR/GITRL axis for cell proliferation and tumorigenicity. **d** Upper panel: The CRL5946 cells were treated with or without 2 μg/mL cisplatin or 7.5 Gy Cs-137 irradiation. Compared with PBS or IgG, cell proliferation was attenuated by GITR neutralized mAb in concentration at 80 μg/mL in nontreated cells (n = 23) or at 20 μg/mL in cisplatin or Cs-137 treated cells (n = 32 for each group). Bottom panel: NOD/SCID mice were injected with nonpretreated, cisplatin-pretreated (2ug/ml for 24 h), or Cs-137 irradiation-pretreated (7.5 Gy, 24 h before injection) cells (n = 8 each treatment group) for 24 days. All mice received intraperitoneal injection of PBS, 100 μg IgG, or anti-GITR on day 0 and day 7. Spheres were decreased in anti-GITR-injected group, whereas PBS- or IgG-injected group did not. All data shown represent the mean ± standard deviation; *P < 0.05; **P < 0.01; ***P < 0.0001 vs PBS or IgG group, determined by one-way ANOVA analysis of variance with Bonferroni post-hoc test.

expression of GITRL was associated with better survival (Fig. 6e). These data in patients treated with the SMART approach confirm that the GITRL/GITR-mediated pathway can have a detrimental impact on survival in nonepithelioid MPM.

## Discussion
The treatment of MPM remains controversial. While patients with epithelioid mesothelioma often appear to obtain benefit from an aggressive approach with multimodality therapy, the outcome of patients with nonepithelioid mesothelioma is uniformly poor and standard therapy with chemotherapy, radiation, and surgery provides limited relief to these patients[4]. Hence, a better understanding of the mechanisms of resistance to standard therapy in nonepithelioid mesothelioma is critical to improve outcome in these patients. Evidence shows that nonepithelioid mesotheliomas are associated with a greater number of markers of epithelial-to-mesenchymal transition (EMT), possibly through epigenetic dysregulation. However, currently no target has been identified to specifically target nonepithelioid mesothelioma[17].

This study is the first to evaluate the role of GITR/GITRL in mesothelioma and to demonstrate that blocking the GITR/GITRL pathway may provide a specific benefit in nonepithelioid mesothelioma.

According to American Type Culture Collection (ATCC), the three human mesothelioma cell lines (CRL5820, CRL5915, and CRL5946) belong to the epithelioid subtype histologically. However, they showed very different morphology in vitro. Histological evaluation of the xenografts generated by these three human mesothelioma cell lines demonstrated that the CRL5820 was epithelioid, CRL5915 biphasic, and CRL5946 sarcomatoid. This observation suggests that cell lines may progressively lose their original characteristics and transform during in vitro culture[16,18,19].

The CRL5946 possesses a higher growth rate, higher resistance to chemo- or radiotherapy, which is consistent with the clinical observation that sarcomatoid subtype is more refractory to conventional therapy and associated with poorer prognosis[3,4,17]. Meanwhile, we demonstrated that the CRL5946 cells also had

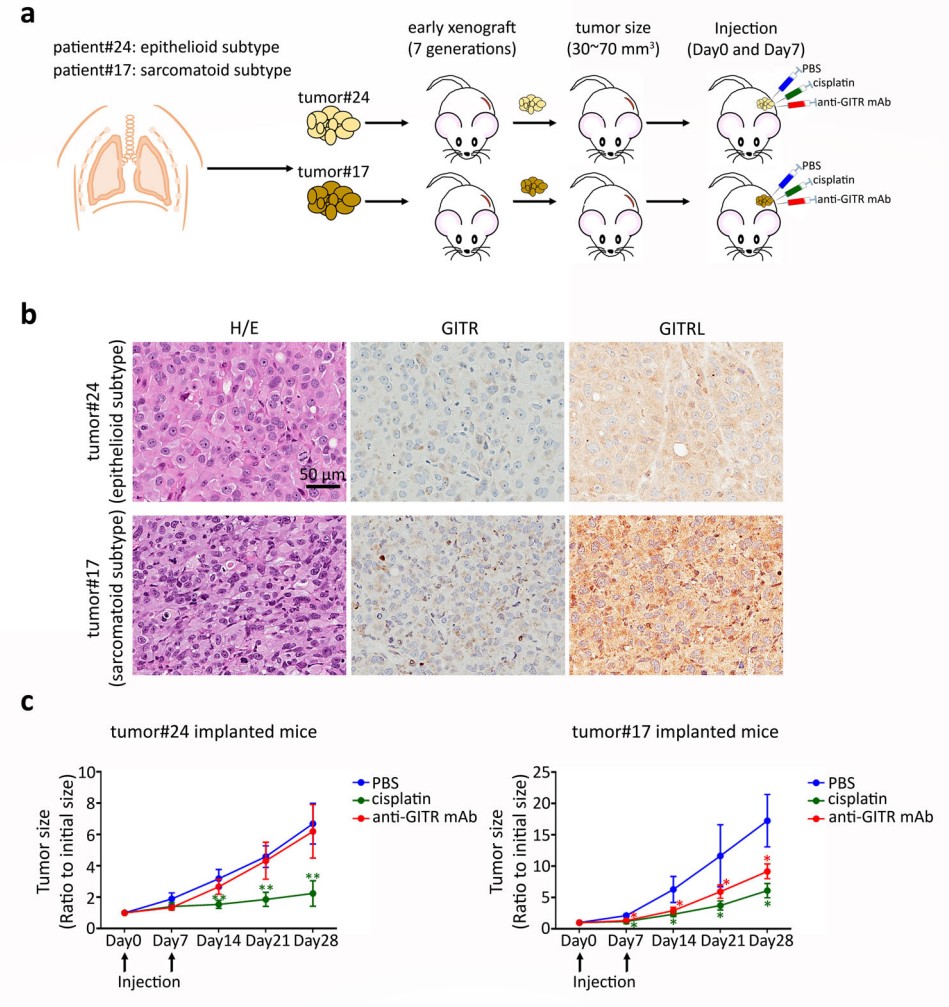

**Fig. 5 Interruption of GITR/GITRL signaling in patient-derived sarcomatoid xenograft inhibits tumor growth. a** Tumours were collected from clinical mesothelioma patients with epithelioid MPM subtype and sarcomatoid MPM subtype and subcutaneously implanted into NOD/SCID mice on right flank region to generate xenografted mice. Mice with xenografted tumours that were the size of 30–70 mm³ received intraperitoneal PBS on day 0 and 7, 5 mg/kg cisplatin on day 0 and 7, or 400 μg anti-GITR mAb on day 0 and 200 μg on day 7. **b** Representative images of H&E stain for morphologic analysis and immunohistochemistry staining for GITRL and GITR expression. **c** Tumor size was decreased in sarcomatoid xenograft mice treated with cisplatin ($n = 4$) or anti-GITR mAb ($n = 4$) and decreased in epithelioid xenograft mice treated with cisplatin ($n = 5$), whereas anti-GITR mAb group ($n = 4$) had no effect on tumor size compared with PBS group ($n = 4$ of sarcomatoid xenograft, n = 6 of epithelioid xenograft). *$P < 0.05$; **$P < 0.01$ vs PBS injected group, determined by Mann–Whitney test.

higher expression of both GITR and GITRL than the other two mesothelioma cell lines (CRL5820, CRL5915). Flow cytometry analysis showed that GITR+ and GITRL+ cells were two distinct single positive subpopulations in CRL5946. We purified the GITR+, GITRL+, and GITR−GITRL− cells using the magnetic beads isolation method. The Principal Component Analysis (PCA) disclosed that the GITR+ and GITRL+ cells have more similarity in their gene expression profiles between each other than with GITR−GITRL− cells sharing pathways related to cell proliferation and tumorigenesis with downregulation of pathways related to cell apoptosis.

Post-translational protein modification is an important regulatory mechanism of cellular proteins associated with carcinogenesis. Upregulation of dolichyl-diphosphooligosaccharide biosynthesis causes aberrant protein N-glycosylation, which was shown to be associated with squamous carcinoma development[20,21]. In other studies, dysregulation of sumoylation may induce cell proliferation, apoptosis resistance, and metastasis

leading to accelerate carcinogenesis[22,23]. In this study, our transcriptome analysis showed GITRL+ cells have an increase in the dolichyl-diphosphooligosaccharide biosynthesis pathway, but a decrease in sumoylation pathway. Furthermore, we found an increase in UVC-induced MAPK and neuregulin signaling in GITRL+ cells, compared to GITRL−GITR− cells. The activation of MAPK signaling triggered by inflammatory cytokines or inflammatory microenvironment leads to cell proliferation; meanwhile, engagement of neuregulin with ERBB3 or ERBB4 receptors activate PI3K/Akt survival pathway. Both of these pathways enhance cell proliferation and cell survival[24–26]. The transcriptome profiling of GITR+ cells revealed several intracellular signaling that may modulate cell cycle for cell growth and downregulation of tumor suppressor such as the p53 signaling. Cancer cells have upregulated glycolysis compared to normal cells, which can lead to the assumption that oxidative phosphorylation (OXPHOS) is universally downregulated in cancer. This is indeed observed for many cancers; however, in some

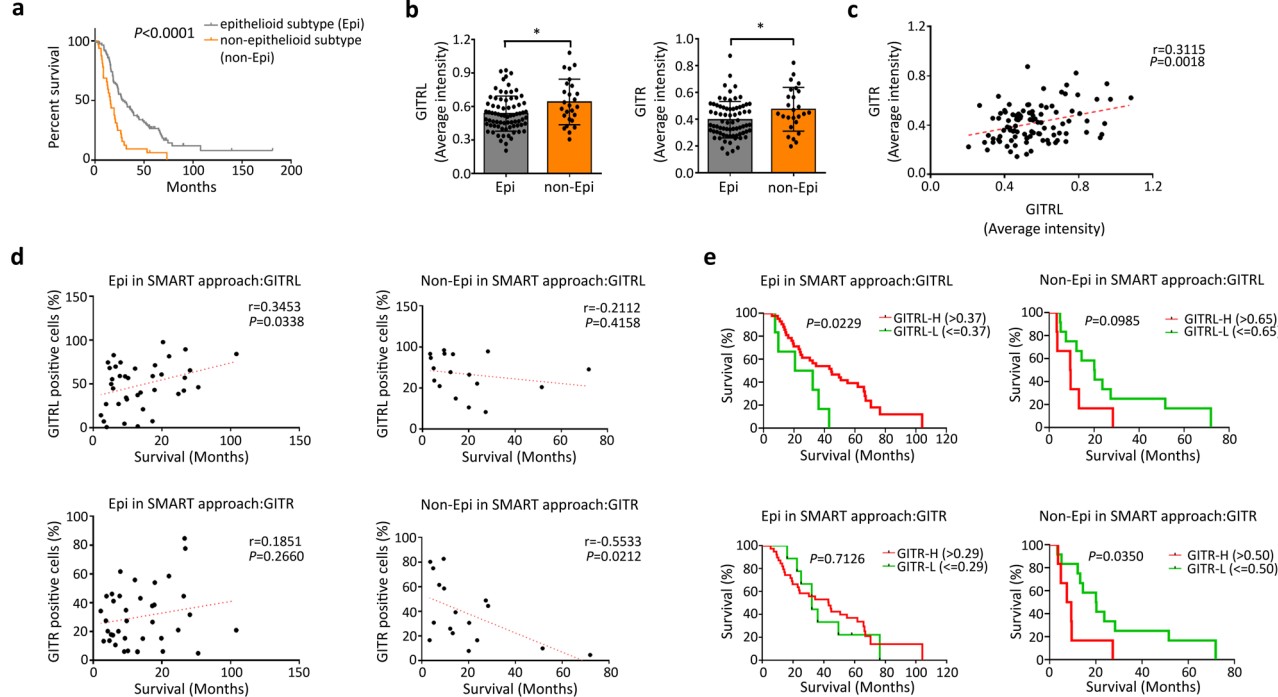

**Fig. 6 GITR/GITRL were highly expressed in nonepithelioid MPM and negatively correlated with survival after the SMART approach. a** Kaplan–Meier survival analysis of all MPM patients demonstrates that patients with epithelioid subtype (Epi.) had better survival than those with nonepithelioid subtype (non-Epi.) with a median survival of 28.6 (Epi.) versus 15.9 months (non-Epi.) (P < 0.0001). **b** GITR and GITRL levels by immunohistochemistry in epithelioid and nonepithelioid MPM; immunohistochemistry results were quantified by Definiens Tissuestudio 4.0 software. *P < 0.05, determined by Mann–Whitney U test. **c** Correlation between expression of GITR and GITRL (p = 0.0018, r = 0.3115). **d** Risk analysis of patients with epithelioid and nonepithelioid subtype who died from mesothelioma after the SMART approach to determine the correlation between GITRL and GITR and length of survival. **e** Kaplan–Meier survival analysis in nonepithelioid MPM showed that high expression of GITRL and GITR was associated with worse survival after SMART. In epithelioid MPM, high expression of GITRL was associated with better survival after SMART.

cancers, this assumption is being challenged by an increasing body of evidence to suggest that mitochondrial metabolism is not impaired. Such cancers include leukemias, pancreatic ductal adenocarcinoma, high OXPHOS subtype melanoma, and endometrial carcinoma[27,28]. Our data on MPM cells revealed that OXPHOS pathway is more highly upregulated in GITR+ cells than in GITRL–GITR– cells. By quantifying these signaling networks within GITRL+ and GITR+ cells, the functional implications can further inform the development of novel biologic therapies. All together, these results demonstrate that the GITR+ and GITRL+ cells have more expression of genes related to aggressiveness, proliferation, and stemness-like traits than GITR–GITRL– cells.

In the functional analysis, GITR+ and GITRL+ cells can form more mesospheres in vitro and in vivo, which is consistent with the results of the IPA analysis. Based on the hypothesis that GITR and GITRL expression can activate an autocrine signal, we used a monoclonal antibody (N-mAb) to block the GITR/GITRL axis to validate this hypothesis. We observed that the N-mAb could slow down the growth in vitro and diminish the number of mesospheres in vivo. Moreover, blocking of GITR/GITRL pathway also decreased tumor growth in sarcomatoid PDXs (Meso#17). However, the N-mAb did not have an effect in epithelioid PDXs (Meso#24), which may be attributed to their low expression level of both GITR and GITRL. Although stemness is often associated with the quiescent state of cancer cells[29,30], expression of GITR and GITRL in CRL5946 cells seemed to generate higher self-renewal capacity, tumorigenicity and higher proliferating rate than the nonexpressing subtype. Similar findings in proteins associated with cancer stemness have been

published for AF4/FMR2 family member 4 (AFF4) and sex-determining region Y box2 (SOX2) in head and neck squamous cell carcinoma[31,32]. We may infer that the role of GITR and GITRL in self-renewal and proliferation may largely depend on cell type and microenvironment. Nevertheless, the GITR/GITRL axis may be an important novel target in treating nonepithelioid MPM.

Epithelial-mesenchymal-transition (EMT) is a physiopathological process by which epithelial cells acquire the properties and shapes of mesenchymal cells. Accumulating evidence suggests that EMT programs the cancer cells to acquire aggressive cell-biological traits, therapeutic resistance, and stem-cell-like characteristics, a trend associated with metastatic dissemination and poor clinical outcomes[33–35]. This would support and explain the higher growth rate and therapeutic resistance to chemotherapy and radiotherapy observed in the sarcomatoid subtype of mesothelioma. Hmeljak et al.[36] also discovered that in comparison to the epithelioid subtype, nonepithelioid MPMs have significantly higher EMT score as well as greater expression of the member of tumor necrosis factor/receptor superfamily, OX40 and OX40L[36]. Blum et al. also showed that TNFSF18 (gene expressing GITRL) was specifically expressed in mesothelioma with large sarcomatoid component, supporting the critical role of this gene in sarcomatoid MPMs[37]. Sarcomatoid mesothelioma contains high proportion of Th2 cells, which can induce expression of GITRL[36]. The contribution of GITR/GITRL in MPM needs to be further elucidated by knockin and knockout experiments with transfection of GITR/GITRL gene into GITR–GITRL– mesothelioma cells and CRISPR knockout of the GITR/GITRL genes in highly expressing mesothelioma cells.

Traditionally, two models have been proposed to explain the stem-cell-like heterogeneity within individual tumor: the hierarchy model and the stochastic model. In the hierarchy model, there are biologically distinct classes of cells with different functional abilities and behavior. In contrast, the stochastic model postulates that cancer cells exist in distinct phenotypic states that differ in functional attributes in which some phenotypes have stemness-like properties[38,39]. In our data with CRL5946, the cancer cells displayed distinct equilibria in proportions of GITR+, GITRL+, and GITR−GITRL− cells. This demonstrates that the heterogeneity within CRL5946 originated from the stochastic interconversion between different phenotypic states.

In our cohort study of 117 MPM patients, epithelioid subtype patients have significantly better survival than nonepithelioid subtype patients, which is consistent with the literature[1,40]. The expression of both GITR and GITRL is higher in nonepithelioid subtype than in epithelioid MPMs, which can explain the rapid cellular growth, higher resistance to chemotherapy or radiotherapy, and higher tumor-initiating capacity of nonepithelioid MPMs. While analyzing survival, we included only patients treated with the SMART approach to homogenize the therapy. We found that GITR and GITRL expressions were associated with worse survival in nonepithelioid mesothelioma, supporting the negative impact of GITR and GITRL expression on outcome in this subtype. In contrast, GITRL expression was associated with better outcome in epithelioid mesothelioma treated with SMART, potentially suggesting that the immune system has a more important role in epithelioid mesothelioma than in nonepithelioid mesothelioma. This possibility is supported by Fear et al.[41] who demonstrated that GITR agonist decreased tumor growth in a murine subcutaneous mesothelioma model and by our recent study demonstrating that greater number of CD8+ T cells were associated with better survival in epithelioid mesothelioma but not in biphasic mesothelioma after the SMART approach[42]. Our in vivo experiments performed in NOD/SCID mice provided the advantage to analyze the intrinsic mechanism of tumor progression in the absence of an immune response supporting the importance of the GITR/GITRL axis in tumor progression in nonepithelioid MPM, while GITR/GITRL had no impact in epithelioid MPM in the absence of an immune response.

The benefit of neutralizing GITR in patients with nonepithelioid mesothelioma might be compromised by interference with the GITR-mediated effector T cell activation and the resulting antitumor immune response. One potential mechanism to overcome this limitation is to enhance the proportion of activated effector CD4+ T cells expressing GITR in the tumor microenvironment before administering GITR agonist to overcome the potential mechanisms of resistance generated by tumor cells. Work in our preclinical mouse model using immunocompetent mice and murine cell lines have shown that the combination of GITR agonist with a short course of radiation and interleukin-15 superagonist was synergistic by boosting the upregulation and activation of effector CD4+ T cells and cytotoxic CD8+ T cells before the administration of GITR agonist. This approach could potentially be an effective strategy in nonepithelial MPM.

These observations carry important information for the development of immunotherapy. Indeed, antibody targeting the immune system may have opposite impact on tumor cells and immune cells. For instance, anti-GITR agonist has been successfully developed and is entering phase I clinical trials. While this target may be beneficial in some immunoreactive tumors by activating effector T cells and depleting regulatory T cells, they may be associated with an absence of response in less immunogenic tumor or even potentially lead to worse outcome by stimulating cancer cells expressing high level of GITR or GITRL. Hence, specific knowledge of the intrinsic effect of immunotherapeutic target on cancer cells will be increasingly critical as these targets are being explored.

## Methods

**Human malignant mesothelioma cell lines and mice.** Human mesothelioma cell lines, NCI-H28 [H28] (ATCC® CRL5820™), NCI-H2052 [H2052] (ATCC® CRL5915™), and NCI-H2452 [H2452] (ATCC® CRL5946™), were purchased from ATCC (American Type Culture Collection). The expression of cytokeratins 5/6 (CK5/6), pan-cytokerain (AE1/AE3), and calretinin on these three cell lines were confirmed by western Blot. Human mesothelioma cells were cultured in RPMI-1640 medium supplemented with 10% FBS (fetal bovine serum) and 1% penicillin and streptomycin and maintained in a humidified atmosphere containing 5% $CO_2$ at 37 °C. Eight to ten-week-old female immunocompromised mice, NOD.CB17-Prkdc$^{scid}$/J strain (NOD/SCID) were purchased from Jackson Laboratories. The animals were housed in microisolator cages, five per cage, in a 12 h light/dark cycle with sterile water and rodent food supply. Animal care and experiments were performed in accordance with institutional and Canadian Institute of Health guidelines. All animal experiments were approved by the Animal Research Ethics Board at the Toronto General Research Institute (University of Toronto, Toronto, CA).

**Determination of histological feature of the human mesothelioma cell lines**. Human mesothelioma cells were resuspended in PBS and then injected into NOD/SCID mice (For each mouse: $6 \times 10^6$ cells in 0.5 ml PBS) either subcutaneously or intraperitoneally. For subcutaneously injected mice, they were sacrificed while the obvious palpable subcutaneous tumor developed; for the intraperitoneally injected mice, they were sacrificed and the intraperitoneal mesospheres were collected by lavage with cold PBS. Then these xenograft tumors were kept in 10% formalin then sent to pathologist for histological confirmation.

**Radiotherapy and chemotherapy on mesothelioma cell lines.** Human mesothelioma cells (CRL5820, CRL5915, and CRL5946) were resuspended in culture medium and treated with cisplatin (Onco-Tain™ Hospira, UK) overnight, or exposed dosage of γ-ray radiation by Cs-137 Gamma Cell Irradiator-40 (Atomic Energy of Canada Ltd., Ottawa, Canada) at a dose rate of ~100 cGy/min. Then they were seeded onto culture plates. After 24 h, the culture medium was removed and then washed twice with warm PBS to remove the dead floating cells and cisplatin and then replaced by fresh culture medium.

The short-term and long-term cytotoxicity of cisplatin and γ-rays on different human mesothelioma cell lines was determined by counting viable cells and the number of colonies, respectively. Human mesothelioma cells (CRL5820, CRL5915, and CRL5946) were resuspended in culture medium and treated with cisplatin 2 µg/ml overnight or γ-ray radiation 7.5 Gy and then seeded onto six-well culture plates. We replaced the culture medium 24 h later.

For short-term cytotoxicity determination, we initially loaded $10^5$ cells onto each well. After 4 days of incubation, we washed twice with warm PBS and resuspended the attached cells in medium. Then the number of viable cells was counted by Cell Counter (Vi-CELL XR, BECKMAN COULTER). Trypan Blue Dye was used to discriminate dead cells from live cells. Cell viability was determined by obtaining the total cells count, live cells and dead cells.

For long-term cytotoxicity determination, we initially loaded 2000 cells onto each well. After 14 days of incubation, we removed the medium then washed twice with cold PBS. The plates were then stained with methylene blue and number of positive colonies (>50 cells) was counted.

**Western blot.** Cultured of treated and nontreated human mesothelioma cell lines (CRL5820, CRL5915, and CRL5946) were washed twice by ice-cold PBS and then lysed in RIPA-Buffer (ThermoFisher Scientific Inc.) containing Protease Inhibitor Cocktail (Sigma–Aldrich Co.) for 15 min on ice. Then all cells were scratched down and transferred to Eppendorf tubes, which was then put on a rotator in 4 °C fridge to keep mixing and digesting for another 20 min. Cellular debris was removed by centrifugation (12,000 rpm, 4 °C, 15 min). Protein concentration was determined by Protein Assay Dye Reagent Concentrate (Bio-Rad Laboratories Inc.). Equal amounts of the protein lysates were separated by SDS-PAGE and then transferred onto nitrocellulose membranes. The membranes were blocked with TBS-Tween containing 4% BSA. After washing with TBS-Tween, the membranes were incubated overnight at 4 °C with the following primary antibodies: rabbit monoclonal anti-human GITRL antibody (Clone:EPR20583, Abcam plc.) (1:1000 in blocking buffer); mouse monoclonal anti-human GITR antibody (Clone:2H4, SIGMA-ALDRICH Co.) (2.5ug/ml) and anti- β -Actin antibody (rabbit anti-human, Bio-Legend Inc.). After washing with TBS-Tween, the blotted membranes were incubated with horseradish peroxidase-conjugated anti-rabbit IgG (Clone:4064, Biolegend Inc.) and anti-mouse IgG antibodies (Clone:4053, Biolegend Inc.) for 40 min at room temperature, respectively. Signals from immunoreactive bands were visualized by fluorography using an ECL reagent (Pierce). The intensity of individual immunoblots was quantified using ImageJ Image program.

**RNA extraction and real-time reverse transcription PCR**. The RNA was extracted from treated and nontreated human mesothelioma cell lines (CRL5820, CRL5915, and CRL5946) treated overnight by using QIAzol Lysis Reagent (QIA-GEN), and RNeasy Microarray Tissue Mini Kit (QIAGEN). cDNA was synthesized with High Capacity cDNA Reverse Transcription Kit (ThermoFisher Scientific) on a C1000 Touch™ Thermal Cycler (BIORAD) following manufacturer's protocols. Regular PCR was done to establish reverse transcription PCR (RT-PCR) standards of all targets genes including GITRL, GITR, and housekeeping gene HPRT1. DNA fragments were obtained from regular PCR on a C1000 Touch™ Thermal Cycler (Bio-Rad). A probe-based real-time PCR approaches for quantitative measurement of targets genes was carried out on the CFX384 Touch real-time PCR detection system (BIO-RAD). PCR composed of 20× PrimePCR Probe Assay, 2× SsoAd-vanced™ Universal Supermix, 2 ul 25 ng/ul cDNA × 45 cycles. The FAM-labeled Probes of PrimePCR Probe Assay of housekeeping gene and all target genes were purchased from BIO-RAD Laboratories, Inc.

**Characterization of GITRL and GITR on CRL5946 cells by fluorescent microscopy stain**. CRL5946 cells were seeded on eight-well culture slides (Corning Life Sciences) with each well $10^4$ cells and treated with cisplatin 2 ug/ml or γ-ray radiation 7.5 Gy. After incubating for 96 h, the medium was removed and the slide was washed twice with cold PBS. The viable cells were fixed with 4% paraformaldehyde for 10 min. The slides were stained with anti-human GITRL antibody (Clone:109101, R&D SYSTEMS), anti-mouse IgG Antibody-AF488 conjugated (Invitrogen), anti-human GITR antibody-PE conjugated (Clone#621, Biolegend Inc.), and DAPI (Cell Signaling) following the commercial instructions. The fluorescence images of whole sides were captured by Nikon A1R+ system, which is built onto a Nikon Eclipse TI-E inverted microscope with Nikon NIS Elements C for acquisition and analysis of images.

**Determination of GITR and GITRL expression of CRL5946 cells by flow cytometry**. CRL5946 cells were detached and resuspended in FACs buffer. Cells were stained for 10 min at 4 °C with a CD16/CD32 Fc block (Biolegend Inc.) and then a combination of the following human-specific antibodies: GITRL-APC conjugated (Clone#109101, R&D Systems co.) and GITR-PE conjugated (Clone#621, BioLegend Inc.). All samples were then washed twice with FACS buffer and fixed with 2% Paraformaldehyde for at least 30 min. BD LSR II flow cytometer (BD Biosciences) and FlowJo V10 software (FlowJo LLC) was then used to analyze the expression of GITR and GITRL on MPM cells.

**Sorting GITR+ GITRL−, GITR−GITRL+, and GITR−GITRL− cells from CRL5946 cells to analyze the gene expression profile**. Cultured CRL5946 cells were detached from culture plate and resuspended in EasySep buffer (STEMCELL Tech.). The sequential sorting of GITR−GITRL+, GITR + GITRL− and GITR −GITRL− was done by using human-specific antibodies: GITRL-APC conjugated (Clone#109101, R&D Systems co.) and GITR-PE conjugated (Clone#621, BioLegend Inc.) and EasySep™ Human APC Positive Selection Kit and EasySep™ Human PE Positive Selection Kit following the manufacturer's instructions. The RNA extraction was performed as previously described. The microarray analysis was done by Human Gene 2.0 ST Array. The data were analyzed by Transcriptome Analysis Console (TAC) Software (ThermoFisher Inc.). The up- and down-regulated genes in GITR + GITRL− and GITR−GITRL+ cells compared to GITR −GITRL− cells ($p < 0.05$, FDR-P < 0.1) were uploaded to Ingenuity Pathway Analysis (IPA) Software (QIAGEN Co. Ltd.) for pathway and functional analysis.

**In vitro and in vivo determination of proliferation rate and tumorigenicity of CRL5946 cells**. The malignant mesothelioma cancer cell line, CRL5946 was used for in vitro and in vivo cell proliferation and tumorigenicity test. The cancer cells were suspended in culture medium then seeding onto 10 cm culture dishes with each dish containing $6 × 10^6$ cells. Then those cells were divided into three groups, each of them received no treatment, cisplatin 2 ug/ml, and Cs-137 7.5 Gy radiation, respectively. Then those cells were incubated in 37 °C incubator with 5% $CO_2$ and humidified atmosphere for 24 h. The dead floating cells were wash out with warm PBS. The attached cells were trypsinized and resuspended in culture medium.

In the in vitro settings, they were then seeded onto 96-wells flat-bottom culture plate (Corning Life Sciences) with 2500 cells in each well. In each group, we treated the cancer cells with PBS, monoclonal Mouse IgG1 (clone # 11711 R, R&D Systems Co.), or neutralizing monoclonal anti-GITR antibody (clone # 110416, R&D Systems Co.). After another 3 days of incubation, the number of viable cancer cells was determined by MTT (3-(4,5-dimethylthiazol-2-yl)2,5-diphenyl tetrazolium bromide) cleavage assay. The value of spectrophotometrical absorbance at 540 nm using ELISA reader represented the number of viable cancer cells.

In the in vivo settings, intraperitoneal injection of the NOD/SCID mice model was used as the platform to conduct the in vivo xenografting experiments. The resuspended pretreated MPM cancer cells were injected into each mice's peritoneum with the $6 × 10^6$ viable cells for each mouse (Fig. S9). Then those mice were divided into three groups: PBS, IgG, and monoclonal anti-GITR antibody. The PBS, IgG, and monoclonal anti-GITR antibody were injected intraperitoneally on Day 0 (100 ug), and Day 7 (100 ug). All mice were euthanized during Day 24. The peritoneum was irrigated with the same amount of cold PBS buffer in each

mouse. The PBS buffer was collected and filtered with the 70 µm cell strainer. The tumor spheres that remained on the mesh were collected and measured under the microscope.

**In vivo determination of proliferation rate of patient-derived Xenografting MPMs**. Patient-Derived Xenografts (PDX) were established from human malignant pleural mesothelioma previously established by our team (16). Two PDXs (#Meso-17, a sarcomatoid subtype, and #Meso-24, an epithelioid subtype) were randomly picked and implanted in right flank subcutaneously of NOD/SCID mice. At the start, 1–2 mm in size PDXs were implanted subcutaneously in the right flank of 8–10-week-old female NOD/SCID mice. After 2–3 weeks, when the subcutaneous tumor grew up to 4–5 mm in diameter they were divided into three groups as control, cisplatin, and monoclonal anti-GITR antibody. The cisplatin and monoclonal anti-GITR antibody were all injected intraperitoneally on Day 0 (monoclonal anti-GITR antibody: 400 ug, Cisplatin 5 mg/Kg) and Day 7 (monoclonal anti-GITR antibody: 200 ug, Cisplatin 5 mg/kg). Tumors size was measured once a week from Day 0. The tumor dimensions were measured using microcaliper and the volume was calculated by the formula: $v = πab^2/6$ (a: longest length, b: perpendicular width). The mice were humanely euthanized when tumor volume hit 1500 $mm^3$ or showed signs of ulcerations as per institutional ethics protocol.

**Collection of clinical data of MPM patients**. The Formalin-fixed paraffin-embedded MPM tissues from 117 MPM patients who received surgery at our institute from November 2003 to October 2015 were used for further IHC staining to evaluate the expression of GITR and GITRL. Tumor tissue and clinical information including pathological type and survival from date of diagnosis and from date of treatment initiation in SMART patients were collected under University Health Network Institutional Research Ethics Board (REB) approved protocols (No.19–5122 and 19–5858). A waiver of informed consent was granted by the REB. The SMART trial registration number from ClinicalTrials.gov was NCT00797719.

**Chromogenic IHC stain and quantification of GITR/GITRL expression of MPMs tissues from MPM patients**. All tissues were collected under an institutional ethical board-approved protocol (19–5122). Formalin-fixed paraffin-embedded blocks of selected patients were cut into serial sections of 4 µm depth and stained with hematoxylin and eosin (H&E), GITR and GITRL. For the H&E staining we followed the protocols from our institutions with some modifications. For the GITR and GITRL staining, the endogenous peroxidase activity was blocked with 3% hydrogen peroxide. Antigen retrieval or unmasking procedure was applied by Tris-EDTA (PH9.0). Serum block was applied according to the instructions of ImmPress species-specific HRP kit (Vector Laboratories Inc., Burlingame, CA). Primary antibodies against GITR (Polyclonal, ThermoFisher Inc.), GITRL (Polyclonal, ThermoFisher Inc.) were incubated overnight. Then apply ImmPress reagents as per kit instructions and develop color by using DAB kit (DAKO). Sections were mounted with MM 24 Leica mounting medium (Cat#3801120) after counterstaining lightly with Mayer's Hematoxylin.

Digital whole slide scans were prepared by scanning in the Hematoxylin and Eosin (H&E) stained slide, and the IHC stained GITR and GITRL stained slides (stained on serial sections to the H&E image). The pathologist identified regions of interest corresponding to the tumor bed in the H&E image, and these regions of interest were transferred from the H&E over to the IHC stained slides, which were aligned to be in the same orientation as the H&E. Following the targeted identification of the tumor regions within each of these slides, the image and region of interest were loaded into Definiens TissueStudio software v4.0 (Definiens Inc, Munich Germany). Within the software, the IHC images were subjected to stain separation to identify hematoxylin and DAB staining intensity; and cells were identified using a nucleus detection algorithm, followed by a cell simulation algorithm that expanded from the nucleus by 2 microns, to simulate a cytoplasmic and membrane area. The intensity of DAB was measured in the entire "cell" region (nucleus and cytoplasm), and thresholds were set to discriminate positive from negative staining, utilizing control slides, as well as regions of high and low intensity on the tissue sections, in consultation with the pathologist. The readout of the quantification was reported in terms of the percent of cells that were positive as well as the average intensity of GITR and GITRL normalized to the control slides.

**Statistics and reproducibility**. Data were processed using Graphpad Prism 6, FlowJO V10 software, and Bio-Rad CFX manger V3.1 software, and all data are presented as the mean ± standard deviation for continuous responses. The unpaired two-tailed Student t-test was used to analyze data from two groups. The one-way ANOVA was used to analyze data from more than two groups. The ROC (Receiver-Operating Characteristic) curve was plotted to determine if a continuous variant could predict the results and the ideal threshold to separate these different results. The Kaplan–Meier survival analysis and log-rank test was used to compare the difference of survival between two specific groups of patients. A p value of less than 0.05 was considered as statically significant for all comparisons. Results have been presented as mean ± SD. *$P < 0.05$; **$P < 0.01$; ***$P < 0.001$ in all figures.

**Reporting summary**. Further information on research design is available in the Nature Research Reporting Summary linked to this article.

## Data availability
The authors confirm that the data supporting the findings of this study are available within the article and its supplementary materials. The clinical results are not publicly available due to ethical restrictions. The microarray data in this manuscript are deposited in ArrayExpress and available at fgsubs #511432. Source Data can be found in Supplementary Data 1. All other data are available from the corresponding author on reasonable request.

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

## Acknowledgements
This work was supported, in part, by the Princess Margaret Hospital Foundation (PMHF), the Mesothelioma Applied Research Foundation and the Canadian Mesothelioma Foundation. M.D.P. hold the Canada Mesothelioma Foundation Professorship in Mesothelioma Research at University Health Network. We thank Ms. Kitty Liu for her editing of this manuscript.

## Author contributions
All authors were involved in drafting the article for important intellectual content, and all authors approved the final version to be published. M.D.P. and M.C. had full access to all of the data in the study and take responsibility for the integrity of the data and the accuracy of the data analysis. Study conception and design: M.C., L.W., M.D.P. Acquisition of data: M.C., L.W., Z.Y., T.D.M., M.C., Y.Z., M.K., J.M. Analysis and interpretation of data: M.C., L.W., M.D.P.

## Competing interests
M.D.P. received personal fees from Astra-Zeneca, Bayer, and Actelion outside of the submitted work. Other authors declare that they have no competing interests.
