## [Peer Review File · Communications Biology]

Reviewers' Comments:

Reviewer #1:

Remarks to the Author:

Goal, rationale and major findings

This manuscript reports investigations on the potential of modulation of GITR/GITRL pathway to overcome resistance to chemo- (CT) and radio- (RT) therapies in malignant pleural mesothelioma (MPM). In front of the limited MPM patients' survival, studies dealing with resistance of MPM to current therapies are strongly needed. Focusing on GITR/GITRL is an original and novel approach in MPM.

To achieve their goal, the authors carried out several studies on GITR/GITRL axis and role of chemo/RT, in vitro with use of malignant mesothelial cells and in vivo with patient-derived mesothelioma xenografts in immune compromised mice. They also used human MPM tumors previously investigated in a therapeutic approach with Surgery for Mesothelioma After Radiation Therapy (SMART) developed by one of the authors (MdP). Therapeutic agents were cisplatin, a currently used molecule in MPM treatment, and Cs-137 irradiation in reference to the SMART study.

With use of neutralizing anti-GITR mAb, the authors found that the GITR/GITRL axis promotes cell growth in the mesothelioma cell type used (defined as sarcomatoid type, the most aggressive form of MPM), in a context of CS or RT pre-treatment, and that patients' survival was negatively correlated with GITR and GITRL expression in patients receiving pre-RT.

General comments

- This study shows some well-done investigations and interesting findings.
- o The manuscript is well written. The work is organized in logical steps.
- Expression of GITR/GITRL in mesothelioma cells and human mesothelioma tumors.
- Effect of CT and RT on GITR/GITRL expression on in vitro and in vivo systems.
- Isolation of mesothelioma cell subpopulations in one cell line, CRL-5946, demonstrating differential expression of GITR and/or GITRL and gene expression profile of these subpopulations.
- Response of the subpopulations of mesothelioma cells to CS and RT, effect on cell viability.
- Tumor potential of the subpopulations and role of CS and RT on tumor growth, and effect of post-treatment with anti-GTR mAb.
- Role of the GITR/GITRL axis in proliferation of mesothelioma cells.
- Role of anti-GTR mAb on patient-derived xenografted (PDX) tumor growth in mice.
- Dependence of patients' survival according to GITR and GITRL expression in tumor cells.
- However, some specific comments can be done from the reading of this manuscript, addressing several questions and points to clarify.

Specific comments

•Some questions arise during the various stages of the study

o Characterization of ATCC mesothelioma cells is limited. While CRL-5946 (H2452) was obtained from a non-smoker with mesothelioma characterized as epithelial CRL-5820 (H28) and CRL-5915 (H2052) were obtained from pleural effusions derived from metastatic site. This brings two questions:

- Except CRL-5946, were these cells of mesothelial origin ?
- CRL-5946 would deserves to be better characterized. Morphological characterization is insufficient and use of epithelial biomarkers could be helpful to confirm the morphological status. Expression of cadherins and repressors of its expression would be helpful. Additionally the authors may refer to literature data. (see for instance PMID: 21983934 or PMID: 28090191). Cell morphology does not rightly defines the cell status and the molecular heterogeneity recently under investigations underlines the importance of other parameters to account for the cell entity and its response to anti-cancer agents.

o On pages 4 and 5, lines 98-103, the sentences are a little misleading:

- "In CRL5946, dose-dependent cytotoxicity test for cisplatin and Cs-137 irradiation showed that

2µg/ml of cisplatin and 7.5 Gy of Cs-137 irradiation caused 70-80 % cell death (Figure S1)” and “When treated with 2 µg/mL cisplatin or 7.5 Gy Cs-137 irradiation, we found more prominent proliferation and colony formation in CRL5946 than in CRL5820 or CRL5915 (Figure 1D and E). The writing should be modified.

o In the cytotoxicity test for cisplatin and Cs-137 irradiation the authors suggest a role of GITR/GITRL, based on counting the number of cells. However, as non-treated CRL5946 cells have the highest number of cells after 4-day culture, in comparison with CRL5820 or CRL5915, it does not seem illogical that the number of cells after treatment, at equivalent dose and time, would be higher than in the other cell lines.

o In vitro/in vivo experiments with CRL5946. The amount of viable cells inoculated should be specified. From Materials and Methods and Legend to Figure 4, it seems that 6x10⁶ cells, either untreated, CS- or RT-treated were inoculated after 24h treatment. In these conditions, what was the number of viable cells inoculated in immune compromised mice?

o In the SMART approach:

- Patients were pre-treated with radiotherapy. If samples are available from histological diagnosis and extrapleural pneumonectomy, could the authors report if GITR and GITRL expression was modified by the treatment ? Can they qualify/quantify the consequences of pre-treatment ?
- Additionally, it would be interesting to know the immune microenvironment of tumor cells in both situations, regarding the role of GITR MODULATION in T cells regulation.

•Some points should be clarified or specified.

o In Figure 1, how was viability measured ?

o The idea of separating CRL-5946 mesothelioma cells according to their GITR and GITRL content was interesting. From Figure 1C it seems that CRL-5946 GITR+/GITRL+ may be also present. Could the authors comment ?

o Both GITR and GITRL are well expressed in CRL5946 in Figure 1C, but not in untreated CRL5946 cells in Figure 2B. This should be clarified.

•More generally

o Data on GITR/GITRL checkpoint activity and control of mesothelioma cells growth. A combination of immune checkpoints, such as PD-1, may be an immunotherapeutic option (see for instance PMID: 31036879). PD-L1 blocking has been used in human MPM (see for instance PMID: 32154179), and several combinations, including GITR-Ab in mesothelioma tumor model in mice (see for instance PMID: 30288361). Could the authors provide information on the activity of other checkpoints in the tested cells ?

o It would be of interest to perform single cell analyses.

o Importantly, one can ask whether histology, as reported here, is the appropriate parameter for mesothelioma cells and MPM identity. Regarding the numerous morphological MPM subtypes, and current findings of mesothelioma molecular heterogeneity, a more precise characterization is needed for a better prediction of the effect of targeted therapies. As the authors studied gene expression of mesothelioma cells, could they consult available public transcriptomic databases to improve the mesothelioma cells identity ?

It is known that there are epithelioid and sarcomatoid histological subtypes, but also molecular subclasses of mesotheliomas (see for instance PMID: 32676358, PMID: 26928227, PMID: 30322867), and a genomic continuum that combined epithelioid and sarcomatoid components (see for instance PMID: 30902996). A molecular characterization would be more pertinent and reliable in a context of targeted therapy.

o Could the authors comment on designing anti-GITR-based immunotherapy in the context of autocrine proliferation of mesothelioma cells and immune tumor microenvironment [see for instance PMID: 31867277 PMID: 29601534 and recent preprint (note that it is not yet reviewed) <https://www.medrxiv.org/content/10.1101/2020.08.14.20174789v1.full.pdf>].

o Discussion on stem cells could be shortened as it is not the focus of the study.

Reviewer #2:

Remarks to the Author:

The authors report on a new resistance mechanism in non-epithelioid mesothelioma based on overexpression of GITR/GITRL. Differences in the GITR/GITRL axis between epithelioid and non-

epithelioid mesothelioma are nicely elaborated based on in vitro, pre-clinical and clinical assessments. The potential benefit of a novel treatment option employing neutralizing GITR antibodies has been demonstrated in vitro and in mouse models and appear to show also promise in the clinical setting by demonstrating a connection of the expression levels of GITR and GITRL with a poor prognosis in non-epithelioid mesothelioma patients. The benefit of neutralizing GITR in patients with non-epithelioid mesothelioma might be compromised by interference with the GITR-mediated T cell activation (and the resulting anti-tumor response); this aspect has been covered in the discussion (in particular that due to the use of NOD/SCID mice potential contributions of the immune system can not be assessed properly).

In summary, the paper is well-written, concise, and the reported results support the authors conclusions. Experimental data are novel, well presented and are adequately supported by statistical evaluation. The presented data are of interest to the community.

Specific comments to the data presentation:

- 1) Figure 5C: Addition of figures with actual tumor sizes (not normalized) to the supplement would be appreciated and would help to judge e.g. homogeneity at randomization. Furthermore, additional plots with individual tumor-growth curves for each group would be appreciated and would add substantial information (in light of the low number of mice per group). Were there any early drop-outs/dead mice?
- 2) Figure 6B: Although the quantitation of the staining is described in detail in the materials and methods section, it is not clear what the numbers in the scale do mean. Is the average intensity normalized to control slides? The materials and methods section mentions that data on % positive cells were also obtained. This data set might be added to the supplement.
- 3) Figure 6D&E: Since numbers of available non-epithelioid mesothelioma tumors from the SMART approach is rather low it would be interesting to have this data representation for all 33 tumors as supplemental material.

Specific comment to the discussion:

- 4) A paper describing the efficacy GITR agonism in a mesothelioma mouse model is missing in the discussion (Fear et al. 2018, Oncoimmunology).

There are a few discrepancies / errors which should be resolved:

- 5) In the results section (page 9, line 216) a median survival of 15 months in non-epithelioid subtype is mentioned which does not match the according number in Figure 6 (legend), namely 16 months (15.85).
- 6) In the discussion (page 15, line 348) it is mentioned that GITR and GITRL expression is associated with better outcome in epithelioid mesothelioma. This is correct for GITRL, however Figure 6E demonstrates that GITR expression has no effect on outcome.
- 7) There is a discrepancy with respect to dosing of the neutralizing anti-GITR antibody in the PDX model. The material and methods section (page 21, line 503) states doses of 400 + 200 µg whereas the Figure legend 5 (page 33, line 769) indicates doses of 400 + 400 µg.
- 8) Reference 45 (page 15, line 353) is missing in the reference list, which comprises 40 references.

Answer to Reviewer #1:

We would like to thank the reviewer for taking the time to review our work and for the constructive comments. The manuscript has been modified according to the suggestions.

Goal, rationale and major findings

This manuscript reports investigations on the potential of modulation of GITR/GITRL pathway to overcome resistance to chemo- (CT) and radio- (RT) therapies in malignant pleural mesothelioma (MPM). In front of the limited MPM patients' survival, studies dealing with resistance of MPM to current therapies are strongly needed. Focusing on GITR/GITRL is an original and novel approach in MPM.

To achieve their goal, the authors carried out several studies on GITR/GITRL axis and role of chemo/RT, in vitro with use of malignant mesothelial cells and in vivo with patient-derived mesothelioma xenografts in immune compromised mice. They also used human MPM tumors previously investigated in a therapeutic approach with Surgery for Mesothelioma After Radiation Therapy (SMART) developed by one of the authors (MdP). Therapeutic agents were cisplatin, a currently used molecule in MPM treatment, and Cs-137 irradiation in reference to the SMART study.

With use of neutralizing anti-GITR mAb, the authors found that the GITR/GITRL axis promotes cell growth in the mesothelioma cell type used (defined as sarcomatoid type, the most aggressive form of MPM), in a context of CS or RT pre-treatment, and that patients' survival was negatively correlated with GITR and GITRL expression in patients receiving pre-RT.

General comments

- This study shows some well-done investigations and interesting findings.
- o The manuscript is well written. The work is organized in logical steps.
- Expression of GITR/GITRL in mesothelioma cells and human mesothelioma tumors.
- Effect of CT and RT on GITR/GITRL expression on in vitro and in vivo systems.
- Isolation of mesothelioma cell subpopulations in one cell line, CRL-5946, demonstrating differential expression of GITR and/or GITRL and gene expression profile of these subpopulations.
- Response of the subpopulations of mesothelioma cells to CS and RT, effect on cell viability.
- Tumor potential of the subpopulations and role of CS and RT on tumor growth, and effect of post-treatment with anti-GTR mAb.
- Role of the GITR/GITRL axis in proliferation of mesothelioma cells.
- Role of anti-GTR mAb on patient-derived xenografted (PDX) tumor growth in mice.
- Dependence of patients' survival according to GITR and GITRL expression in tumor cells.
- However, some specific comments can be done from the reading of this manuscript, addressing several questions and points to clarify.

Thank you for the comments.

Specific comments

• *Some questions arise during the various stages of the study*

o Characterization of ATCC mesothelioma cells is limited. While CRL-5946 (H2452) was obtained from a non-smoker with mesothelioma characterized as epithelial CRL-5820 (H28) and CRL-5915 (H2052) were obtained from pleural effusions derived from metastatic site. This brings two questions:

• Except CRL-5946, were these cells of mesothelial origin?

Response: In order to confirm the origin of the human cell lines, we did immunoblotting of the three cell lines for calretinin, CK5/6 and pancytokeratin (AE1/AE3). The results show that each cell line expresses calretinin, CK5/6 and pancytokeratin (AE1/AE3). These findings were included in the material and methods section: “The expression of cytokeratins 5/6 (CK5/6), pancytokeratin (AE1/AE3), and calretinin on these 3 cell lines were confirmed by Western Blot.” (page 17, lines 399-400)

• CRL-5946 would deserve to be better characterized. Morphological characterization is insufficient and use of epithelial biomarkers could be helpful to confirm the morphological status. Expression of cadherins and repressors of its expression would be helpful. Additionally, the authors may refer to literature data. (see for instance PMID: 21983934 or PMID: 28090191). Cell morphology does not rightly define the cell status and the molecular heterogeneity recently under investigation underlines the importance of other parameters to account for the cell entity and its response to anti-cancer agents.

Response: The expression of mesenchymal and epithelial markers were checked for CRL5946 using epithelial markers (ZO-1, E-cadherin) and mesenchymal markers (N-cadherin, β -catenin, Vimentin, Slug, and Snail).

These results show that CRL5946 express very low level of ZO-1 and does not express E-cadherin, whereas mesenchymal markers are mostly expressed. All together the morphology and EMT markers demonstrate that CRL5946 are sarcomatoid-like cells. This finding is consistent with the calretinin expression showed above, which is also implicated in cell proliferation and EMT transition in mesothelioma (PMID: 30555628, PMID: 28285878). Calretinin is expressed on CRL5946 and CRL5915 but has limited expression on CRL5820.

The manuscript was modified to emphasize these findings: “Using molecular markers for mesenchymal characteristics (calretinin, N-cadherin, β -catenin, vimentin, Slug and Snail), we confirmed that CRL5946 and CRL5915 expressed mesenchymal markers supporting their sarcomatoid component.” (page 4, lines 91-93)

o On pages 4 and 5, lines 98-103, the sentences are a little misleading:
 • “In CRL5946, dose-dependent cytotoxicity test for cisplatin and Cs-137 irradiation showed that 2 μ g/ml of cisplatin and 7.5 Gy of Cs-137 irradiation caused 70-80 % cell death (Figure S1)” and “When treated with 2 μ g/mL cisplatin or 7.5 Gy Cs-137 irradiation, we found more prominent proliferation and colony formation in CRL5946 than in CRL5820 or CRL5915 (Figure 1D and E). The writing should be modified.

Response: Thank you for pointing this out. We have revised the sentences in our manuscript to be clearer: “In CRL5946, dose-dependent cytotoxicity test for cisplatin and Cs-137 irradiation showed that 2 μ g/ml of cisplatin and 7.5 Gy of Cs-137 irradiation caused 70-80 % cell death (Figure S1). Thus, we used these dosages to treat mesothelioma cell lines in further experiments. The proliferation rate was higher in CRL5946 than in CRL5820 or CRL5915 at baseline (Figure 1D). After treatment with 2 μ g/mL cisplatin or 7.5 Gy Cs-137 irradiation, the proliferation rate was decreased across all 3 cells lines, but remained similarly superior in CRL5946 (Figure 1D). In addition, after treatment with the same dosage of cisplatin or Cs-137 irradiation, we found more colony formation in CRL5946 than in CRL5820 or CRL5915 (Figure 1E).” (page 5, lines 100-108).

o In the cytotoxicity test for cisplatin and Cs-137 irradiation the authors suggest a role of GITR/GITRL, based on counting the number of cells. However, as non-treated CRL5946 cells have the highest number of cells after 4-day culture, in

comparison with CRL5820 or CRL5915, it does not seem illogical that the number of cells after treatment, at equivalent dose and time, would be higher than in the other cell lines.

Response: This analysis demonstrates that the proliferation rate was similarly affected by chemo and irradiation, hence the ability to proliferate remains superior in CRL5946 compared to CRL5915 and CRL5820. The sentence was modified to specify this point: “The proliferation rate was higher in CRL5946 than in CRL5820 or CRL5915 at baseline (Figure 1D). After treatment with 2 µg/mL cisplatin or 7.5 Gy Cs-137 irradiation, the proliferation rate was decreased across all 3 cells lines, but remained similarly superior in CRL5946 (Figure 1D).” (page 5, line 103-108)

o In vitro/in vivo experiments with CRL5946. The amount of viable cells inoculated should be specified. From Materials and Methods and Legend to Figure 4, it seems that 6x10⁶ cells, either untreated, CS- or RT-treated were inoculated after 24h treatment. In these conditions, what was the number of viable cells inoculated in immune compromised mice?

Response: In vitro/ in vivo experiments, we inoculated 6x10⁶ cells into each dish and divided them into three groups (untreated, cisplatin 2 ug/ml treated, Cs-137 7.5 Gy treated) 24 hours before conducting the experiment. For each group, we prepared 10 dishes. On the day of the experiment, we resuspended all dishes of each group into one conical centrifuge tube. Then we determined the concentration of viable cells and injected 6x10⁶ viable cells into immune-compromised mice peritoneally. The simplified illustration is as follows:

This figure was included as a supplemental figure in the manuscript for clarification (Fig S9).

o In the SMART approach:

• Patients were pre-treated with radiotherapy. If samples are available from histological diagnosis and extrapleural pneumonectomy, could the authors report if GITR and GITRL expression was modified by the treatment? Can they

qualify/quantify the consequences of pre-treatment ?

• Additionally, it would be interesting to know the immune microenvironment of tumor cells in both situations, regarding the role of GITR MODULATION in T cells regulation.

Response: Unfortunately, we do not have the pre-radiation samples for all patients in the SMART trial. However, our new trial (NCT04028570), which was initiated about a year ago, is designed to address this question. Samples are collected before radiation and at the time of surgery after radiation and separated according to the dose of radiation (background dose and boost dose). The samples are processed for scRNA-seq and CyTOF. Our preliminary data demonstrate that GITR is upregulated on T cells and NK cells after radiation, particularly with higher dose of radiation (boost).

•Some points should be clarified or specified.

o In Figure 1, how was viability measured?

Response: The cell counter that we used is “Vi-CELL XR Cell Viability Analyzer” from BECKMAN COULTER Inc. which used Trypan Blue Dye to discriminate dead cells from live cells. So the output data was expressed as total number of cell counted, with the number of live cells and dead cells, which was used to determine cell viability. This point was clarified in the material and methods section: “Trypan Blue Dye was used to discriminate dead cells from live cells. Cell viability was determined by obtaining the total cells count, live cells and dead cells.” (page 18, lines 432-433)

o The idea of separating CRL-5946 mesothelioma cells according to their GITR and GITRL content was interesting. From Figure 1C it seems that CRL-5946 GITR+/GITRL+ may be also present. Could the authors comment?

Response: The fluorescent image analysis (Fig 2c) showed that GITRL and GITR may be coexpressed on the surface of CRL5946 cells. We further labeled the cells with GITRL-APC and GITR-PE human specific antibodies and analyzed them with flow cytometry for cell fraction analysis. We found the percentage of cell fraction was 1.67% in GITR+ cells, 0.7% in GITRL+ cells, and 0.07% in GITR+/GITRL+ cells (Fig 3A). For enrichment of the cell fractions, we used human-specific GITRL-APC & GITR-PE conjugated antibodies and then EasySep™ Human APC Positive Selection Kit & EasySep™ Human PE Positive Selection Kit following the manufacturer’s instructions and subsequently analyzed the purity of cell fractions with flow cytometry. Although the purity increased by around 20 times in the GITR and GITRL population, the proportion remained similar and, consequently, we can’t exclude that a small fraction of double positive GITR+/GITRL+ cells were present. However, this population remained small compared to the GITR and GITRL population (Fig.3A).

o Both GITR and GITRL are well expressed in CRL5946 in Figure 1C, but not in untreated CRL5946 cells in Figure 2B. This should be clarified.

Response: The western blot in Fig. 1C compared the expression of GITR and GITRL among 3 cell lines. The expression of GITR and GITRL was far lower in CRL5820

and CRL5946. However, in Fig. 2B, the expression of GITR and GITRL was performed in different conditioned for CRL5946. After applying ECL, the time of fluorography for capturing images was much longer in Fig. 1C than in Fig. 2B. Actually, in Fig. 2B untreated CRL5946 group still showed a very weak appearance of GITR and GITRL. In conclusion, the condition of performing western blots in Fig. 1C and Fig. 2B is different, which led to the lack of appearance of GITR and GITRL in the untreated group of CRL5946 cells in Fig. 2B. This point was emphasized in the text: “Note that the time of fluorography for capturing the images was shorter in Fig 2B compared to Fig 1C, which explain the weak appearance in the untreated group.” (page 6, lines 129-131)

•*More generally*

o **Data on GITR/GITRL checkpoint activity and control of mesothelioma cells growth. A combination of immune checkpoints, such as PD-1, may be an immunotherapeutic option (see for instance PMID: 31036879). PD-L1 blocking has been used in human MPM (see for instance PMID: 32154179), and several combinations, including GITR-Ab in mesothelioma tumor model in mice (see for instance PMID: 30288361). Could the authors provide information on the activity of other checkpoints in the tested cells?**

Response: Immunotherapy is opening new opportunities in the treatment of mesothelioma. However, in contrast to lung cancer and melanoma, monotherapy with PD-1 or PD-L1 inhibitor has had benefit in a minority of patients with mesothelioma only and a large randomized trial of CTLA-4 inhibitor alone in second line treatment was negative (PMID: 28729154). Therefore, combination therapy will be important. In our experience, single immune checkpoint blockade are associated with minimal response in preclinical models. Our group is studying the impact of combining immunotherapy with chemotherapy or radiation in mesothelioma, which appears to provide encouraging results.

o **It would be of interest to perform single cell analyses.**

Response: Single cell RNA sequencing will be extremely informative to understand the impact of radiation and chemotherapy on GITR expression. As mentioned above, we have initiated a new clinical trial (NCT04028570) and are collecting tumor specimen before and after radiation. The specimen are analyzed by scRNA-seq. We are planning to have the final report on these samples in the course of 2021. These results will help to determine the type of immunotherapy to combine with radiation.

o **Importantly, one can ask whether histology, as reported here, is the appropriate parameter for mesothelioma cells and MPM identity. Regarding the numerous morphological MPM subtypes, and current findings of mesothelioma molecular heterogeneity, a more precise characterization is needed for a better prediction of the effect of targeted therapies. As the authors studied gene expression of mesothelioma cells, could they consult available public transcriptomic databases to improve the mesothelioma cells identity? It is known that there are epithelioid and sarcomatoid histological subtypes, but also molecular subclasses of mesotheliomas (see for instance PMID: 32676358,**

PMID: 26928227, PMID: 30322867), and a genomic continuum that combined epithelioid and sarcomatoid components (see for instance PMID: 30902996). A molecular characterization would be more pertinent and reliable in a context of targeted therapy.

Response: We agree that increasing evidence suggest that molecular characterization of mesothelioma will provide better classification and could become superior to histological evaluation. This work is ongoing and increasingly mesothelioma appears to be a continuous spectrum between the epithelial and sarcomatoid component. We reviewed the molecular characterization based on transcriptomic databases. The most important information in relation to the GITR-GITRL pathway is reported by Blum and colleagues who recently published a new molecular classification based on their own dataset as well as 4 published datasets (Reynies, Bueno, Gordon, Lopez) and the TCGA. They divided mesothelioma into 4 subgroups with C2B tumors expressing the greater proportion of sarcomatoid genes. TNFSF18 (GITRL) was expressed only in C2B tumors. These tumors contained greater proportion of T cells and monocytes and data from the TCGA suggest that these T cells are predominantly Th2. This point was added to the discussion: “Blum et al also showed that TNFSF18 (gene expressing GITRL) was specifically expressed in mesothelioma with large sarcomatoid component, supporting the critical role of this gene in sarcomatoid MPMs (43). Sarcomatoid mesothelioma contains high proportion of Th2 cells, which can induce expression of GITRL.” (page 14, lines 334-337)

o Could the authors comment on designing anti-GITR-based immunotherapy in the context of autocrine proliferation of mesothelioma cells and immune tumor microenvironment [see for instance PMID: 31867277 PMID: 29601534 and recent preprint (note that it is not yet reviewed) <https://www.medrxiv.org/content/10.1101/2020.08.14.20174789v1.full.pdf>]

Response: This is a critical question. GITR-based immunotherapy will require to have abundant cytotoxic T cells as well as possibly NK cells to response to GITR-agonist therapy and overcome the autocrine proliferation of mesothelioma cells. Based on our current investigations, we believe that the use of GITR-agonist will be optimal with a short course of radiation therapy to activate the immune microenvironment and upregulate GITR expression on cytotoxic T cells and NK cells. Using our preclinical models, we demonstrated that the combination of a short course of radiation therapy with IL-15 superagonist and GITR-agonist was very effective by upregulating and activating GITR+ cytotoxic T cells and downregulating regulatory T cells. These series of experiments have recently been accepted for publication in *Science Translational Medicine*. This point was included in the discussion: “The benefit of neutralizing GITR in patients with non-epithelioid mesothelioma might be compromised by interference with the GITR-mediated effector T cell activation and the resulting anti-tumor response. One potential mechanism to overcome this limitation is to enhance the proportion of activated effector CD4+ T cells expressing GITR in the tumor microenvironment before administering GITR agonist to overcome the potential mechanisms of resistance generated by tumor cells. Work in our preclinical mice model using immunocompetent mice and murine cell lines have

shown that the combination of GITR-agonist with a short course of radiation and interleukin-15 superagonist was very effective by boosting the upregulation and activation of effector CD4+ T cells and cytotoxic CD8+ T cells before the administration of GITR agonist. This approach could potentially be an effective strategy in non-epithelial MPM.” (page 16, lines 375-384)

o Discussion on stem cells could be shortened as it is not the focus of the study.

Response: The discussion on stem cells was shortened (page 14-15, lines 343-354).

Answer to Reviewer #2:

We would like to thank the reviewer for taking the time to review our work and for the constructive comments. The manuscript has been modified according to the suggestions.

The authors report on a new resistance mechanism in non-epithelioid mesothelioma based on overexpression of GITR/GITRL. Differences in the GITR/GITRL axis between epithelioid and non-epithelioid mesothelioma are nicely elaborated based on in vitro, pre-clinical and clinical assessments. The potential benefit of a novel treatment option employing neutralizing GITR antibodies has been demonstrated in vitro and in mouse models and appear to show also promise in the clinical setting by demonstrating a connection of the expression levels of GITR and GITRL with a poor prognosis in non-epithelioid mesothelioma patients. The benefit of neutralizing GITR in patients with non-epithelioid mesothelioma might be compromised by interference with the GITR-mediated T cell activation (and the resulting anti-tumor response); this aspect has been covered in the discussion (in particular that due to the use of NOD/SCID mice potential contributions of the immune system can not be assessed properly).

In summary, the paper is well-written, concise, and the reported results support the authors conclusions. Experimental data are novel, well presented and are adequately supported by statistical evaluation. The presented data are of interest to the community.

Response: Thank you for the comments.

Specific comments to the data presentation:

1) Figure 5C: Addition of figures with actual tumor sizes (not normalized) to the supplement would be appreciated and would help to judge e.g. homogeneity at randomization. Furthermore, additional plots with individual tumor-growth curves for each group would be appreciated and would add substantial information (in light of the low number of mice per group). Were there any early drop-outs/dead mice?

Response: This information was provided in supplemental figure (Figure S5). The individual tumor growth is presented in a ratio to initial size (normalized) for the epithelioid xenograft (A) and sarcomatoid xenograft (C). The actual average tumor size as well as the actual individual tumor size for each mouse is then presented for the epithelioid xenograft (B) and for the sarcomatoid xenograft (D). There was no early drop-outs or dead mice within 28 days after initiation of injection of cisplatin or anti-GITR mAb.

2) Figure 6B: Although the quantitation of the staining is described in detail in the materials and methods section, it is not clear what the numbers in the scale do mean. Is the average intensity normalized to control slides? The materials and methods section mention that data on % positive cells were also obtained. This data set might be added to the supplement.

Response: The analysis was performed for both percentage of positive cells and average intensity normalized to the control slides. The material and methods section was modified to describe the quantitation of the staining: “The readout of the quantification was reported in terms of the percent of cells that were positive as well as the average intensity of GITR and GITRL normalized to the control slides.” (page 24, lines 573-574). We included the data on the percentage of positive cells in the supplemental figure (Figure S7).

3) Figure 6D&E: Since numbers of available non-epithelioid mesothelioma tumors from the SMART approach is rather low it would be interesting to have this data representation for all 33 tumors as supplemental material.

Response: Unfortunately, patients treated outside of the SMART protocol were often not monitored in our institution in the long-term and therefore complete follow-up is lacking for these patients. We did update the follow-up for patients in the SMART group and included the new survival graphs in Figure 6, which confirmed the initial results. Long-term follow-up was available for all the SMART patients.

Specific comment to the discussion:

4) A paper describing the efficacy GITR agonism in a mesothelioma mouse model is missing in the discussion (Fear et al. 2018, Oncoimmunology).

Response: This citation was included in the discussion: “In contrast, GITRL expression was associated with better outcome in epithelioid mesothelioma treated with SMART, potentially suggesting that the immune system has a more important role in epithelioid mesothelioma than in non-epithelioid mesothelioma. This possibility is supported by Fear et al (41) who demonstrated that GITR agonist decreased tumor growth in a murine subcutaneous mesothelioma model and our recent study demonstrating that greater number of CD8+ T cells were associated with better survival in epithelioid mesothelioma but not in biphasic mesothelioma after the SMART approach (40).” (page 15, lines 364-370)

There are a few discrepancies / errors which should be resolved:

5) In the results section (page 9, line 216) a median survival of 15 months in non-epithelioid subtype is mentioned which does not match the according number in Figure 6 (legend), namely 16 months (15.85).

Response: Thank you for pointing this out. The error has been corrected in the results section: “The Kaplan-Meier survival analysis of all 117 cases demonstrated a median survival of 26.7 months in epithelioid subtype and 15.9 months in non-epithelioid subtype”. (page 10, lines 226-227)

6) In the discussion (page 15, line 348) it is mentioned that GITR and GITRL expression is associated with better outcome in epithelioid mesothelioma. This is correct for GITRL, however Figure 6E demonstrates that GITR expression has no effect on outcome.

Response: We have modified the text and removed GITR: “In contrast, GITRL expression was associated with better outcome in epithelioid mesothelioma treated with SMART” (page 15, lines 364-365)

7) There is a discrepancy with respect to dosing of the neutralizing anti-GITR antibody in the PDX model. The material and methods section (page 21, line 503) states doses of 400 + 200 µg whereas the Figure legend 5 (page 33, line 769) indicates doses of 400 + 400 µg.

Response: We have corrected the figure legend and mentioned that 400 µg was given on day 0 and 200 µg on day 7. (page 34, lines 806-807)

8) Reference 45 (page 15, line 353) is missing in the reference list, which comprises 40 references.

Response: The missing reference was added, Reference 40: “M. de Perrot, L. Wu, M. Cabanero, J. Y. Perentes, T. D. McKee, L. Donahoe, P. Bradbury, M. Kohno, M. L. Chan, J. Murakami, S. Keshavjee, M. S. Tsao, B. C. J. Cho, Prognostic influence of tumor microenvironment after hypofractionated radiation and surgery for mesothelioma. *J Thorac Cardiovasc Surg* 159, 2082-2091 e2081 (2020).” (page 30)

Reviewers' Comments:

Reviewer #1:

Remarks to the Author:

The authors have, point by point, replied to my comments, providing well developed and convincing answers, and clarifications in some aspects that will improve the validity of their models in the field of mesothelioma.

This manuscript contains original data that can be useful to mesothelioma research and therapeutical approaches.

Reviewer #2:

Remarks to the Author:

All points raised in my review of the initial manuscript were adequately addressed and resolved in the revised manuscript.

One minor point remains: the reference 41 newly introduced to the discussion (Fear et al.) is missing in the "References" section.